# Transcriptional profiling of mESC-derived tendon and fibrocartilage cell fate switch

Deepak A. Kaji[1], Angela M. Montero[1], Roosheel Patel[2] & Alice H. Huang [1✉]

The transcriptional regulators underlying induction and differentiation of dense connective tissues such as tendon and related fibrocartilaginous tissues (meniscus and annulus fibrosus) remain largely unknown. Using an iterative approach informed by developmental cues and single cell RNA sequencing (scRNA-seq), we establish directed differentiation models to generate tendon and fibrocartilage cells from mouse embryonic stem cells (mESCs) by activation of TGFβ and hedgehog pathways, achieving 90% induction efficiency. Transcriptional signatures of the mESC-derived cells recapitulate embryonic tendon and fibrocartilage signatures from the mouse tail. scRNA-seq further identify retinoic acid signaling as a critical regulator of cell fate switch between TGFβ-induced tendon and fibrocartilage lineages. Trajectory analysis by RNA sequencing define transcriptional modules underlying tendon and fibrocartilage fate induction and identify molecules associated with lineage-specific differentiation. Finally, we successfully generate 3-dimensional engineered tissues using these differentiation protocols and show activation of mechanotransduction markers with dynamic tensile loading. These findings provide a serum-free approach to generate tendon and fibrocartilage cells and tissues at high efficiency for modeling development and disease.

---

[1] Department of Orthopaedics, Icahn School of Medicine at Mount Sinai, New York, NY, USA. [2] Department of Microbiology, Icahn School of Medicine at Mount Sinai, New York, NY, USA. ✉email: ah364@cumc.columbia.edu

Tendons are dense, connective tissues whose load-bearing function is enabled by a fibrous extracellular matrix (ECM) composed of aligned type I collagen and other minor components[1–3]. With injury, tendon ECM is replaced by disorganized scar, leading to impaired function. Regenerative strategies for tendon are limited as the basic cell and molecular regulators underlying tendon differentiation remain largely unknown.

The defining transcription factors for tendon cell fate, identified from developmental studies, include *Scx*, *Mkx*, *Egr1*, and *Egr2*. Although *Scx* is the earliest marker for tendon progenitors, it is not required for tendon induction or maintenance[4,5]. Similarly, null mutations in *Mkx*, *Egr1*, and *Egr2* do not result in overt embryonic tendon phenotypes[6–8]. In addition to tendons, *Scx* is also detected in related connective tissues, such as the meniscus and the annulus fibrosus of the intervertebral disc[9,10]. Compared to tendon, these tissues have mixed fibrous and cartilage (fibrocartilage) elements, characterized by the presence of type II collagen and proteoglycans, in addition to type I collagen[11].

To date, the TGFβ pathway remains the primary signaling pathway identified for mammalian tendons, as it is required for tendon induction and maintenance[12,13]. Interestingly, TGFβ also induces the chondrogenic cell fate as it is required for induction of skeletal cartilage in vivo and is frequently used to induce cartilage in vitro[14–17]. Although tendon, fibrocartilage, and cartilage cell fates exist along a continuum and arise from common mesenchymal progenitors[16,18], the transcriptional and molecular signals that regulate the switch between these tissues have not been defined. Large-scale transcriptomic profiling efforts such as ENCODE and single-cell RNA sequencing (scRNA-seq) atlases consistently omit dense connective tissues such as tendons and fibrocartilage[19–21]. Only two transcriptomic studies for embryonic mouse tendon have been carried out using microarray and RNA sequencing (RNA-seq) of sorted ScxGFP cells; however, these analyses bypassed initiating events underlying *Scx* induction[22,23]. Thus, the transcriptional and molecular regulators that govern tendon induction have still not been identified.

Stem cell differentiations are ideal models to investigate the regulators of cell fate and lineage specification. Currently, there are very few protocols for directed differentiation of tendon cells from pluripotent sources[24,25]. These prior work with embryonic stem cells (ESCs) and induced pluripotent stem cells used limited markers to confirm tendon cell fate[25], and resulting induction efficiencies were limited (~6–18% efficiency) or not reported[24,26].

In this study, we established models of tendon and fibrocartilage induction by leveraging developmental signals to differentiate mouse embryonic stem cells (mESCs). Using single-cell RNA sequencing (scRNA-seq), we defined our differentiated *Scx* populations against their relevant in vivo embryonic counterparts. Informed by scRNA-seq, we refined the signaling environment to improve final induction efficiency to ~90% and uncovered retinoic acid as a molecular driver of TGFβ-induced tendon versus fibrocartilage fates. We further profiled temporal trajectories of tendon and fibrocartilage induction using RNA sequencing (RNA-seq) to identify factors regulating induction. Finally, we successfully generated three-dimensional (3D) engineered tissues using these defined media and showed enhancement or maintenance of tendon and fibrocartilage differentiation, respectively, in concert with activation of mechanotransduction pathways in response to dynamic tensile loading. These results represent a comprehensive investigation of tendon and fibrocartilage induction from pluripotent progenitors and establish a model system for studying tendon development and tendon mechanobiology in vitro.

## Results

**Derivation of ScxGFP tendon cells from mESCs by activation of TGFβ and hedgehog signaling.** mESCs derived from mouse blastocysts containing the ScxGFP reporter[9] were aggregated to form embryoid bodies (D0-D2) in the presence of BMP inhibitor LDN-193189, followed by 4 days of paraxial mesoderm (PM) specification using Wnt agonist CHIR and LDN-193189 (D2-D6, Fig. 1a)[27]. For tendon differentiation, we then cultured PM cells with tenogenic induction media containing TGFβ1 for an additional 4 days (D6-D10). FGF2 was used in all conditions throughout tenogenic differentiation (D6-D10) to maintain cell survival and served as baseline control (Base) for tendon induction[27–29]. With FGF2-only, we observed a modest level of ScxGFP expression (15–35%) at D10 (Fig. 1b–e). The addition of TGFβ1 enhanced ScxGFP+ induction efficiency between 40–80% (Fig. 1b–e). Interestingly, we identified two ScxGFP+ populations in the TGFβ1 condition, which we designated ScxGFP high (ScxGFP^hi) and ScxGFP low (ScxGFP^lo). The addition of TGFβ1 increased the ScxGFP^hi population relative to Base (Fig. 1b–d). Since ScxGFP induction was observed in the Base condition, we considered the possibility this may be the result of autocrine TGFβ signaling. We therefore included the TGFβ inhibitor SB-43154 with FGF2, and found that inhibition of TGFβ signaling inhibited ScxGFP^hi induction, but total ScxGFP induction was not significantly affected (Fig. 1b, e).

Despite robust ScxGFP induction, TGFβ1 supplementation resulted in ~50% overall ScxGFP induction efficiency at D10, suggesting ~50% of the culture remained undifferentiated or adopted alternative fates. To improve tenogenesis, we considered other relevant signals involved in tendon induction. Since axial tendons are derived from a subcompartment of the sclerotome termed the syndetome[30], we considered sclerotomal induction signals, such as sonic hedgehog, that may improve induction[31–35]. Using a small molecule smoothened agonist to activate the hedgehog pathway (SAG, D6-D10), we found similar levels of ScxGFP induction between SAG and TGFβ1 conditions at D10 (Fig. 1c–e). However, more ScxGFP^hi cells were observed with TGFβ1 (almost one log fold change higher) compared to SAG. Inhibition of TGFβ signaling with SB-431542 in the SAG condition (D6-D10) resulted in near-complete suppression of ScxGFP at D10 (Fig. 1c). The combination of TGFβ1 + SAG was synergistic, with the highest numbers of total ScxGFP+ cells as well as ScxGFP^hi cells (Fig. 1c–e). Since ScxGFP+ induction by SAG was TGFβ-dependent, it was possible that increased ScxGFP induction was due entirely to extra autocrine release of TGFβ ligands. We therefore tested increasing concentrations of TGFβ1 (10–80 ng/mL) in the absence of SAG, relative to 10 ng/mL TGFβ1 + SAG (D6-D10). While no difference was observed in ScxGFP induction with increasing TGFβ1 concentrations (10–80 ng/mL) at D10, the addition of SAG in combination with the lowest TGFβ1 concentration (10 ng/mL) increased both overall ScxGFP+ and ScxGFP^hi cell populations (Fig. 1f). This data suggests that SAG modifies cell responsiveness to TGFβ1 to enhance ScxGFP induction.

Having established the utility of SAG in enhancing ScxGFP efficiency, we next confirmed differentiation by gene expression analysis of mESC pluripotency genes (*Nanog*, *Oct4*, *Sox2*)[36]. As expected, pluripotency genes were highly expressed in naïve mESCs at D0, but significantly downregulated with paraxial mesoderm and tendon differentiation at D6 and D10, respectively (Supplementary Fig. 1). Expression of *Scx* was dramatically induced only in the D10 tenogenic condition, relative to both

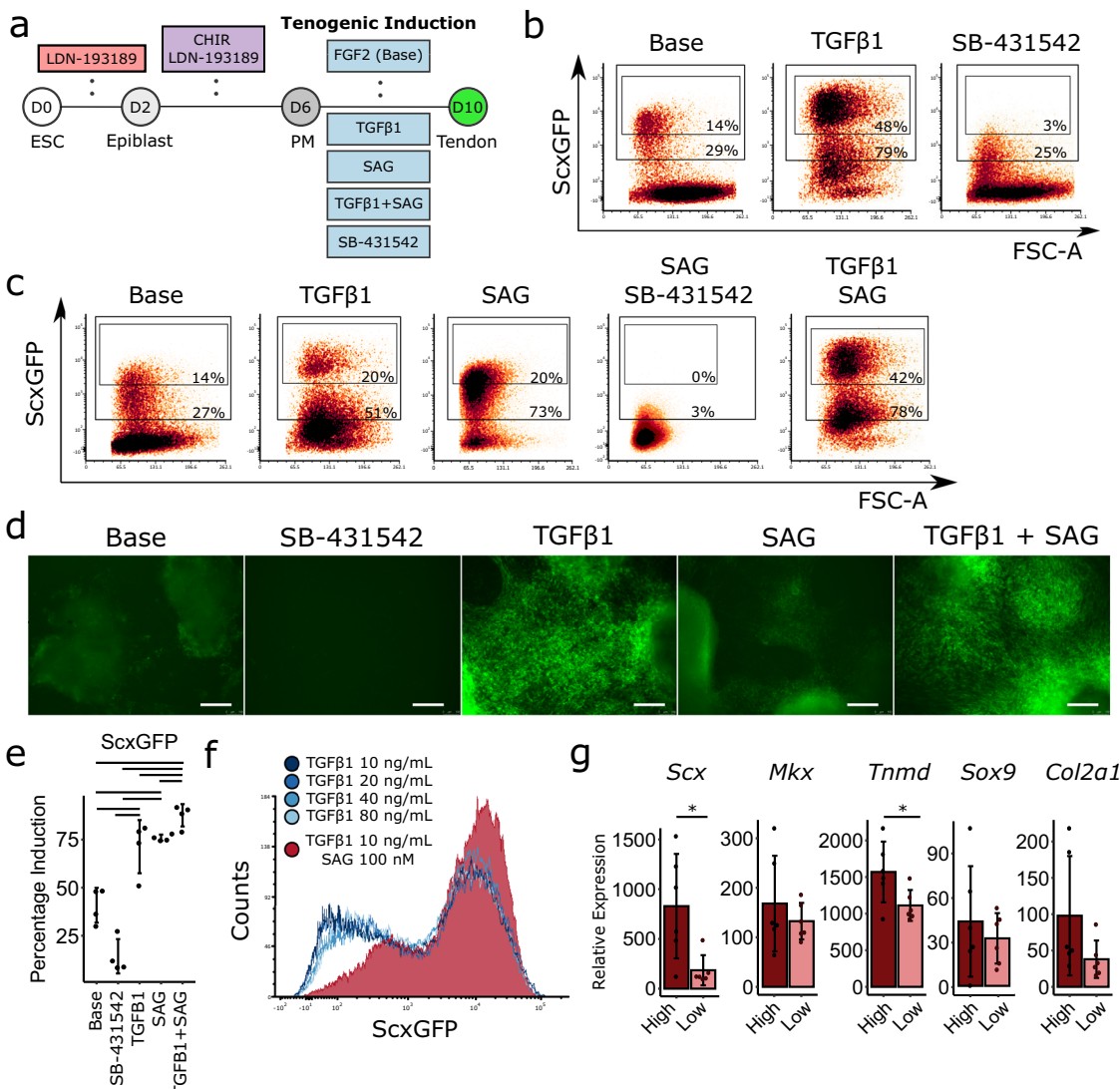

**Fig. 1 Hedgehog activation modifies mESC responsiveness to TGFβ signaling and improves ScxGFP induction. a** Schematic of tendon differentiation protocol. Top row indicates common media components. Bottom row indicates media component variables. **b, c** Flow cytometry plots for ScxGFP induction under tenogenic media conditions. All conditions include FGF2. **d** Representative images of ScxGFP expression for tenogenic media conditions. All conditions include FGF2. All experiments were repeated a minimum of three times. Scale bar: 100 μm. **e** Flow cytometry quantification of ScxGFP induction for tenogenic media conditions ($n = 4$ independent samples, one-way ANOVA with Tukey's post-hoc tests). All experiments were repeated a minimum of three times. Data represented as mean ± SD. **f** Counts histogram showing ScxGFP induction with increasing concentrations of TGFβ1 compared to TGFβ1 + SAG condition. **g** Real-time qPCR quantification of tendon and cartilage genes in sorted ScxGFPhi and ScxGFPlo cells ($n = 6$ independent samples, unpaired two-sided Student's $t$-test). All experiments were repeated a minimum of three times. Data represented as mean ± SD. Bars indicate $p < 0.05$ between groups. *$p < 0.05$. Source data are provided as a source data file.

naive D0 mESCs and D6 PM, confirming tendon lineage specification. To determine whether ScxGFPhi and ScxGFPlo populations represented distinct tendon cell subpopulations, we next performed cell sorting at D10 (TGFβ1 + SAG) and analyzed tendon and cartilage gene expression by qPCR (Fig. 1g). Interestingly, ScxGFPlo cells expressed lower levels of *Scx* and *Tnmd* compared to ScxGFPhi cells, but no significant difference in cartilage markers *Sox9* and *Col2a1* was observed ($p > 0.1$). This data suggests that ScxGFPlo cells may represent a transitional population in the process of turning on ScxGFP and tendon markers.

In addition to hedgehog activation, we also considered BMP signaling, since BMP signaling was previously shown to inhibit *Scx* expression during embryonic chick and mouse

development.[16,37,38] We therefore hypothesized that some level of BMP activation may be modulating tenogenic induction in our cultures and that inhibition of BMP signaling may therefore improve induction. As expected, supplementation with BMP4 resulted in nearly total inhibition of ScxGFP+ cell induction (4%) at D10, consistent with the developmental literature (Supplementary Fig. 2). However, the inclusion of the BMP inhibitor, LDN-193189 in combination with TGFβ1 resulted in reduced ScxGFP induction compared to TGFβ1 treatment alone (27% decrease). Interestingly, while BMP4 treatment resulted in the loss of ScxGFP+ cell induction, the combination of TGFβ1 + BMP4 resulted in only modest reduction of ScxGFP+ cells compared to TGFβ1 alone (13% decrease). Analysis of gene expression by qPCR at D10 showed no differences in *Scx*, *Mkx*, or

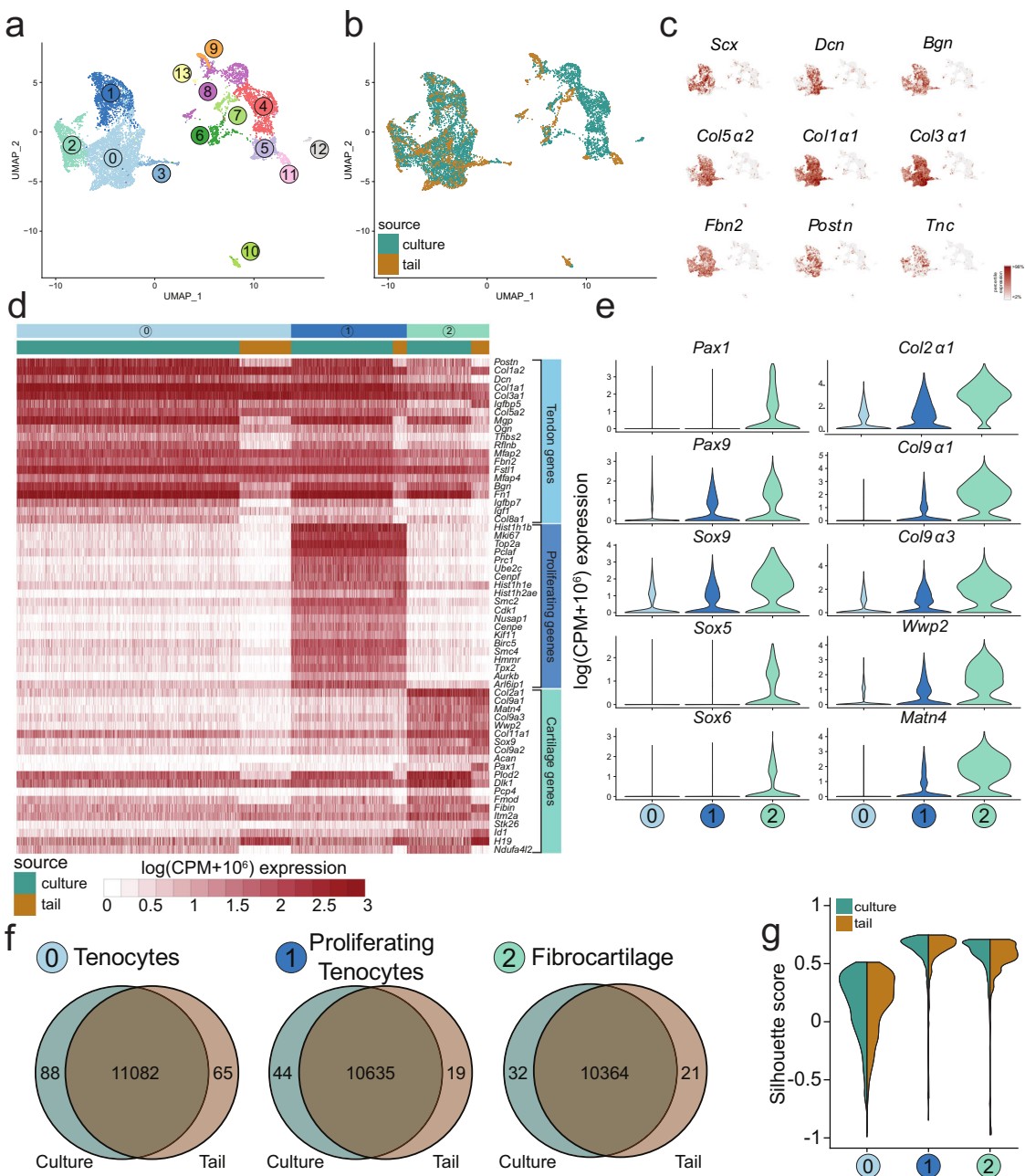

**Fig. 2 Single-cell RNA sequencing of E14.5 tail cells and mESC-derived culture cells reveals tendon and fibrocartilage populations. a**, **b** UMAP of all 14 clusters identified by modularity analysis from the E14.5 tail or the culture. **c** Feature plots of known tendon genes for clusters. **d** Top 20 differentially expressed genes for clusters 0, 1, and 2 visualized by heatmap. **e** Violin plots for transcription factors (left) and proteins (right) involved in cartilage matrix production and homeostasis. **f** Venn diagrams comparing differentially expressed genes between E14.5 tail cells and culture cells in Clusters 0, 1, and 2. **g** Silhouette scores between E14.5 tail cells (blue) and culture cells (brown) in Clusters 0, 1, and 2.

*Tnmd* expression between the TGFβ1, TGFβ1 + BMP4, or TGFβ1 + LDN groups (Supplementary Fig. 2). We further considered that BMP4 treatment may be inducing a chondrogenic phenotype, however, *Sox9* and *Col2a1* expression were also not significantly different between BMP4 alone compared to the TGFβ1-treated groups (Supplementary Fig. 2).

**Single-cell RNA sequencing revealed high transcriptional similarity between D10 mESC-derived *Scx* cells and embryonic E14.5 *Scx* cells**. To fully define the mESC-derived cell populations, we performed scRNA-seq. Using the TGFβ1 + SAG condition which generated the highest induction of ScxGFP+ cells,

we compared the entire culture population at D10 to embryonic mouse tail populations at E14.5. The E14.5 stage was chosen as tendons are fully differentiated[4].

We successfully sequenced 2372 cells isolated from the E14.5 tail (4548 median genes/cell and 83,835 mean reads/cell) and 8382 cells isolated from the D10 mESC differentiation protocol (culture) (2658 median genes/cell and 18,879 mean reads/cell). Using the in vivo embryonic tail cells as our gold standard, we performed unsupervised clustering of E14.5 tail populations, which partitioned cells into 13 clusters (Fig. 2a). Clusters were then annotated based on the expression of select marker genes. Using the annotated tail clusters as the in vivo reference dataset, the cells from the D10 culture were assigned to the clusters using

**Table 1 Percent contribution of E14.5 tail cells and D10 mESC-derived culture cells to each scRNA-seq cluster.**

| Cluster | E14.5 tail cells (% total) | D10 culture cells (% total) |
|---|---|---|
| 0 | 685 (33.4%) | 2964 (37.9%) |
| 1 | 186 (9.1%) | 1353 (17.3%) |
| 2 | 49 (2.4%) | 1088 (13.9%) |
| 3 | 241 (11.7 %) | 855 (10.9%) |
| 4 | 121 (5.9%) | 556 (7.1%) |
| 5 | 71 (3.5%) | 435 (5.6%) |
| 6 | 86 (4.2%) | 257 (3.3%) |
| 7 | 82 (4.0%) | 155 (2.0%) |
| 8 | 157 (7.6%) | 53 (0.7%) |
| 9 | 137 (6.7%) | 2 (0.0%) |
| 10 | 71 (3.5%) | 58 (0.7%) |
| 11 | 128 (6.2%) | 1 (0.0%) |
| 12 | 0 (0.0%) | 43 (0.5%) |
| 13 | 38 (1.8%) | 1 (0.0%) |

a nearest neighbors approach[39]. The combined culture-tail dataset revealed many highly integrated clusters (Fig. 2a, b).

Next, we conducted differential gene expression (DGE) analysis to identify the genes defining each cluster (Supplementary Dataset 1). To identify tendon clusters, we assessed expression of select marker genes including *Scx*. Clusters 0, 1, and 2 all expressed *Scx* as well as other known tendon ECM markers such as *Dcn*, *Bgn*, *Col5α2*, *Col1a1*, *Col3α1*, *Fbn2*, *Postn*, and *Tnc* (Fig. 2c, d)[22,40,41]. Interestingly, while cluster 3 (0.6% of culture cells) did not express *Scx*, it did express all of the other tendon markers consistent with clusters 0, 1, and 2. In total, *Scx*-expressing clusters comprised ~67% of the culture, within the range of ScxGFP induction with TGFβ1 + SAG (Fig. 1e).

While cluster 0 (38% of culture cells) was distinguished primarily by tendon marker expression, cluster 1 (17% of culture cells) was defined by expression of both tendon and proliferation genes (Fig. 2d). Gene set enrichment analysis showed statistically significant cell cycle terms (Supplementary Fig. 3), suggesting this cluster represents proliferating tendon cells. Surprisingly, when we attempted to regress out cell cycle effects, this cluster remained intact. While top-ranked genes (by log fold change) for cluster 1 were genes related to cell proliferation, we also detected significant low-level differences in sclerotomal genes such as *Pax9* and its downstream targets *Sox9* and *Col2α1* (Fig. 2e), suggesting that low-level activation of the sclerotomal program maintains a proliferative, progenitor state in cluster 1 tendon cells. While expression of tendon genes were also detected, cluster 2 (14% of culture) was defined primarily by cartilage genes, such as *Col2α1*, *Col9α1*, *Matn4*, *Col11α1*, *Sox9*, and *Acan* (Fig. 2d, e). The combined expression of tendon and cartilage genes suggested that cluster 2 represents fibrocartilage cells[11]. DGE analysis of culture versus tail cells within clusters 0, 1, and 2 revealed low numbers of differentially expressed genes (Fig. 2f, Supplementary Dataset 2). Further, silhouette analysis of these clusters showed excellent integration between tail and culture (Fig. 2g), indicating successful recapitulation of tendon and fibrocartilage phenotypes comparable to in vivo cells.

In addition to fibrous clusters, gene set enrichment and dot plot analysis also showed the presence of other expected cell types within the embryonic tail, including skin (clusters 4 and 7), blood (clusters 5 and 11), nerves (cluster 9), and muscle (cluster 8) (Supplementary Dataset 1, Supplementary Fig. 3, Supplementary Fig. 4). While larger clusters contained a mixture of tail and culture cells, the smaller clusters tended to be source-specific; for example, clusters 11 and 12 were composed largely of tail-only and culture-only cells, respectively (Table 1). Comparison of percentage composition of culture and tail cells in each cluster revealed that while neural cells comprised ~6% of the tail population, only 0.03% of culture cells were assigned to this cluster. Similarly low percentages of skin (2%) and muscle (0.7%) were observed in the culture population (Table 1), indicating minimal differentiation toward these non-tenogenic lineages using our protocol. For larger non-fibrous clusters (such as cluster 4, representing 7% of the culture population), silhouette analyses also showed that the culture signatures were not well integrated with tail signatures despite assignment to the same cluster (Supplementary Fig. 3). In the case of cluster 4, gene enrichment showed that culture cells expressed genes such as *Nanog*, that were not expressed by tail cells in these clusters (Supplementary Fig. 3). This suggested that the cluster 4 culture cells may represent a small population of pluripotent cells that escaped differentiation.

**Retinoic acid signaling regulates the switch between tendon and fibrocartilage cell fates.** The scRNA-seq results confirmed the presence of differentiated tendon and fibrocartilage cells, however, a significant population of cells adopted or maintained other fates. We hypothesized that this reflected inefficient mESC differentiation to PM based on the composition of cluster 4 culture cells. To identify the signals that may antagonize PM induction, we performed scRNA-seq on PM culture cells at D6. 6695 D6 PM cells were successfully sequenced with 26,589 mean reads/cell and 3170 median genes/cell. Unsupervised clustering of D6 PM cells revealed that the majority of the cells were in a closely grouped supercluster (cluster 0, 2, 3, 4, and 6), while other cells had diverged (clusters 1, 5, 7, and 8, Fig. 3a).

To better define the signaling pathways driving these transcriptional differences, we performed differential gene expression and gene set enrichment analyses and identified the retinoic acid pathway as the most activated pathway at the D6 PM stage. *Rbp4*, a retinol-binding protein, was the most differentially expressed gene in the largest cluster by log fold change (Fig. 3b, Supplementary Dataset 3). We examined other retinoic acid pathway elements and found *Rbp1*, *Rarα*, and cluster-specific expression of *Aldh1a2*. To test the effect of retinoic acid signaling at the PM induction stage, we included a retinoic acid pathway activator (all-trans retinoic acid, ATRA) or inverse agonist (AGN 193109) during the PM differentiation phase (D2-D6). All PM conditions were then cultured with TGFβ1 + SAG from D6-D10 (Fig. 3c). Remarkably, retinoic acid pathway activation during PM induction by ATRA eliminated almost all ScxGFP induction at D10 (Fig. 3d). Treatment with RARγ-specific agonist CD1530 during D2-D6 also showed suppression of ScxGFP, albeit to a lesser extent compared to ATRA (14% vs 5%), which targets all RAR isoforms (Supplementary Fig. 5). In contrast, treatment with AGN 193109 resulted in an increase in ScxGFP+ cells, compared to the Base condition at D10 (Fig. 3d). The failure of ScxGFP induction at D10 despite TGFβ1 + SAG treatment suggested a failure of PM specification at D6 with ATRA activation of retinoic acid signaling. Since retinoic acid signaling is known to maintain pluripotency as well as induce neural differentiation[42–44], we tested these possibilities with ATRA treatment (D2-D6) compared to AGN 193109 treatment. Gene expression analysis at D6 showed downregulation of pluripotency marker *Oct4* in both AGN 193109 and ATRA conditions while epiblast markers *Eomes*[45] and *Fgf5*[45] were suppressed in ATRA only (Supplementary Fig. 5). Analysis of fate markers showed upregulation of paraxial mesoderm marker *Tbx6*[46] with AGN 193109 treatment compared to upregulation of neural marker *Sox1*[47] with ATRA treatment (Supplementary Fig. 5). These results suggested that activation of retinoic acid signaling resulted in differentiation toward a neural fate and therefore loss of cells that

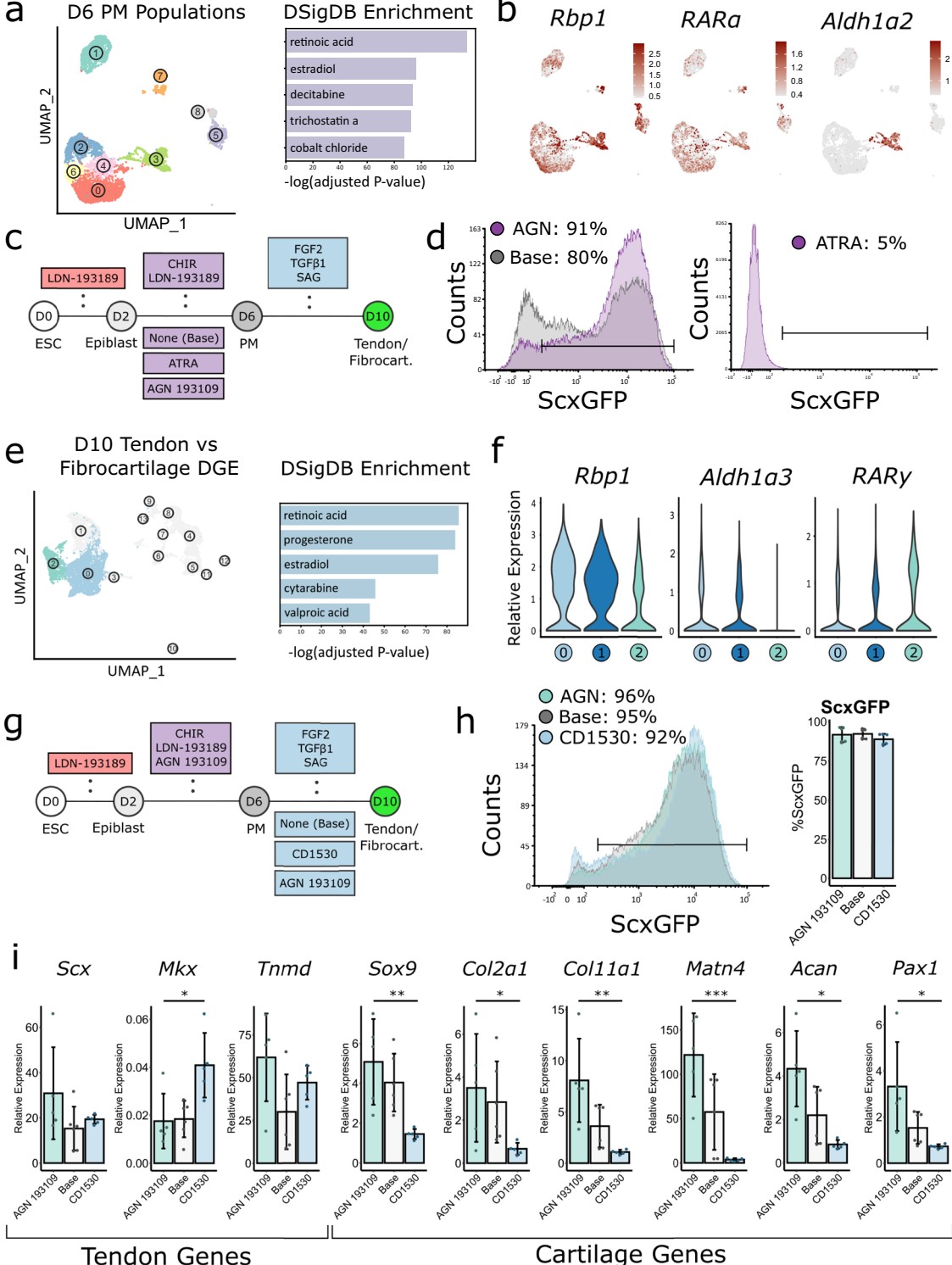

were competent at D6 to undergo subsequent tendon/fibrocartilage differentiation, despite exposure to TGFβ1 + SAG at D6–D10.

Having improved ScxGFP induction efficiency, we next addressed fate specification between tendon and fibrocartilage. To uncover potential molecular drivers regulating the tendon/fibrocartilage fate switch, we performed DGE testing between tendon (cluster 0) and fibrocartilage (cluster 2) cells at D10, using the scRNA-seq dataset collected from the D10 TGFβ1 + SAG condition. Gene set enrichment analysis again identified the retinoic acid pathway, which was the top-ranked term by p-value

(Fig. 3e). We also found that *Rbp1* and *Aldh1α3* were more highly expressed in the tendon clusters relative to the fibrocartilage cluster (Fig. 3f). Conversely, the retinoic acid receptor, *Rarγ*, was upregulated in the fibrocartilage cluster relative to tendon clusters (Fig. 3f). Since the retinoic acid pathways are known to modulate chondrogenesis[48–50], we tested supplementation with the RARγ-specific agonist CD1530 and the inverse agonist AGN 193109, in combination with TGFβ1 + SAG during the tendon induction phase (D6–D10) (Fig. 3g). At D10, ScxGFP induction efficiency was equivalent between all three conditions (~90%, Fig. 3h).

**Fig. 3 Retinoic acid pathway manipulation improves ScxGFP induction efficiency and direct tendon and fibrocartilage fate switch. a** UMAP of clusters from D6 PM. DSigDB gene set enrichment analysis on most differentially expressed genes from each cluster in D6 PM. **b** Feature plots of *Rbp1, RARα, Aldh1α2*. **c** Schematic for differentiation protocol with ATRA, AGN 193109, or no supplementation (Base) during PM induction. Top row indicates common media components. Bottom row indicates media component variables. **d** Counts histograms of ScxGFP expression comparing AGN 193109, ATRA, and Base conditions at D10. **e** UMAP of E14.5 tail cells and D10 culture cells with tendon and fibrocartilage clusters highlighted. DBSigDB gene set enrichment analysis on the most differentially expressed genes in the tendon cluster 0 relative to fibrocartilage cluster 2. **f** Violin plots of *Rbp1, Aldh1α3, RARγ* for Clusters 0, 1, and 2. **g** Schematic for tendon and fibrocartilage differentiation with retinoic acid pathway manipulation using CD1530 and AGN 193109. Top row indicates common media components. Bottom row indicates media component variables **h** Counts histograms of ScxGFP expression and quantification of ScxGFP+ cells at D10 ($n = 4$ independent samples, one-way ANOVA). All experiments were repeated a minimum of three times. Data represented as mean ± SD. **i** Real-time qPCR analysis of tendon and cartilage genes at D10 identified from the single-cell transcriptional signatures. D10 expression is normalized to D6 PM gene expression ($n = 6$ independent samples, unpaired two-sided Student's *t*-test between AGN 193109 and CD1530 conditions). All experiments were repeated a minimum of three times. Data represented as mean ± SD. $*p < 0.05$, $**p < 0.01$, $***p < 0.001$. Source data are provided as a source data file.

Among tendon genes, only *Mkx* expression increased with pathway activation (CD1530), while *Scx* and *Tnmd* were not affected (Fig. 3i). However, treatment with CD1530 drastically reduced cartilage gene expression (Fig. 3i). Using the Mouse Proteome Profiler Array, we further screened and identified enhanced phosphorylation of receptor tyroskine kinases associated with chondrogenesis in the AGN 193109-treated group compared to CD1530, such as PDGFRα, PDGFRβ, and ErbB2 (Supplementary Fig. 6)[51–54]. These data indicate that retinoic acid signaling in the presence of TGFβ1 and SAG regulates the chondrogenic switch between mESC-derived tendon and fibrocartilage fates without adversely affecting the fibrogenic phenotype. These results further suggest that the biological response to retinoic acid signaling is highly context-specific and depends on the competency of the cells experiencing signaling as well as the presence of other signaling pathways (Supplementary Fig. 7).

**Molecular profiling of tendon and fibrocartilage differentiation trajectories by RNA sequencing.** Using these highly efficient differentiation protocols to specify tendon and fibrocartilage cells, we next profiled the molecular dynamics of fate induction for both lineages. RNA-seq was performed on D6 PM as the starting common control and subsequent timepoints (D6.25, 6.75, 7.5, 8.5, and 10) sampled for CD1530 and AGN 193109 conditions (105 M reads/sample, Fig. 4a). To determine the kinetics for tendon and fibrocartilage fate acquisition in the CD1530 and AGN 193109 conditions, respectively, we first examined the tendon, proliferating tendon, and fibrocartilage gene signatures (identified from scRNASeq clusters 0, 1, and 2, respectively) (Supplementary Fig. 8). The proliferating tendon signature was downregulated in both trajectories after D8.5, consistent with terminal differentiation to post-mitotic lineages (Supplementary Fig. 8). This indicates that modulation of retinoic acid signaling pushed differentiation toward cluster 0 "tendon" or cluster 2 "fibrocartilage" phenotypes, without inducing a cluster 1 "proliferating tenocyte" population. Consistent with the expected phenotypes, tendon genes (*Scx, Mkx, Col1a1, Col3a1*) were highly upregulated in both conditions at D10 while cartilage genes (*Col2a1, Acan, Matn4*) were much higher in the AGN 193109 condition and minimally expressed in the CD1530 condition at D10 (Supplementary Fig. 8).

Principal component analysis (PCA) revealed a time dimension captured by PC0, while tendon versus fibrocartilage differentiation was captured by PC1 (Fig. 4b). Interestingly, transcriptional signatures associated with cell fate (PC1) were minimally affected until D8.5, after which both the tendon and fibrocartilage trajectories rapidly upregulated their respective signatures. By D10, tendon and fibrocartilage signatures were strongly activated. Divergence of fate began after D7.5 and increased dramatically between D8.5 and D10 (Fig. 4b). To

identify the genes contributing to this divergence, we identified the genes with the largest contributions to PC0 and PC1 (Supplementary Dataset 4). Among the largest contributors to PC0 were *Postn, Bgn, Col1a2, Col3a1, Col5a2*, and *Tnc*, consistent with the fibrous signature shared by both lineages. Genes that contributed to upward movement along PC1 included *Lox* (tendon marker associated with late embryonic differentiation[55]), and *Igfbp5* (highly expressed in scRNA-seq tendon clusters 0 and 1 and previously identified in embryonic tendon attachment cells[22,56]) (Fig. 2, Supplementary Fig. 8). In contrast, genes that contributed to downward movement along PC1 included *Tgfbi, Col2a1*, and *Vcan* which are established cartilage markers (Supplementary Fig. 8). Interestingly, many genes contributing to PC0 also contributed to PC1 in the upward or downward direction. This suggests that many genes are highly upregulated in both lineages compared to PM, but differentially expressed between tendon and fibrocartilage.

We then performed DGE testing to identify statistically significant genes contributing to each fate trajectory (Figs. 4c, 5). Fate-independent (PC0) and -dependent (PC1) modules of highly correlated genes were determined by performing clustering of differentially expressed genes in separate correlation spaces. Since PC0 represented 79% of the variance of the dataset and tendon and fibrocartilage trajectories did not diverge until D8.5, the fate-independent modules revealed early induction gene patterns common to both cell types. Analysis of fate-independent modules identified multiple genes with intriguing kinetics, including one large module (Module 1) that showed rapid upregulation in gene expression between D6 and D7.5 before stabilizing (Fig. 4c). Transcription factors within this module included known markers of the syndetome (*Scx*) and sclerotome (*Pax1* and *Pax9*), as well as other transcription factors previously unassociated with tendon or cartilage differentiation (Fig. 4c, Supplementary Dataset 4). Interestingly, a separate module with a similar pattern (Module 4) contained the transcription factor *Sox9*. Genes within this module were upregulated more quickly than Module 1 genes, before subsequently downregulating. In this dataset, *Sox9* and *Pax1* were higher in AGN 193109 vs CD1530, supporting fate divergence (Fig. 4c). *Tbx6* and *Cdx1* were both expressed in PM and contributed to Module 3. Rapid downregulation of these genes is consistent with efficient differentiation. Lastly, Module 2 was only transiently expressed, as shown by representative markers *Myog* and *Tal1* (Fig. 4c, Supplementary Dataset 4). Although *Myog* is a key factor of muscle differentiation[57,58], its expression was also previously identified in embryonic tendon cells of the limb[23].

Analysis of fate-dependent modules revealed tendon modules whose expression peaked at D8.5 (Module 5) and D7.5 (Module 6) in the CD1530 condition, with little change or decrease in the AGN 193109 condition (Fig. 5, Supplementary Dataset 4).

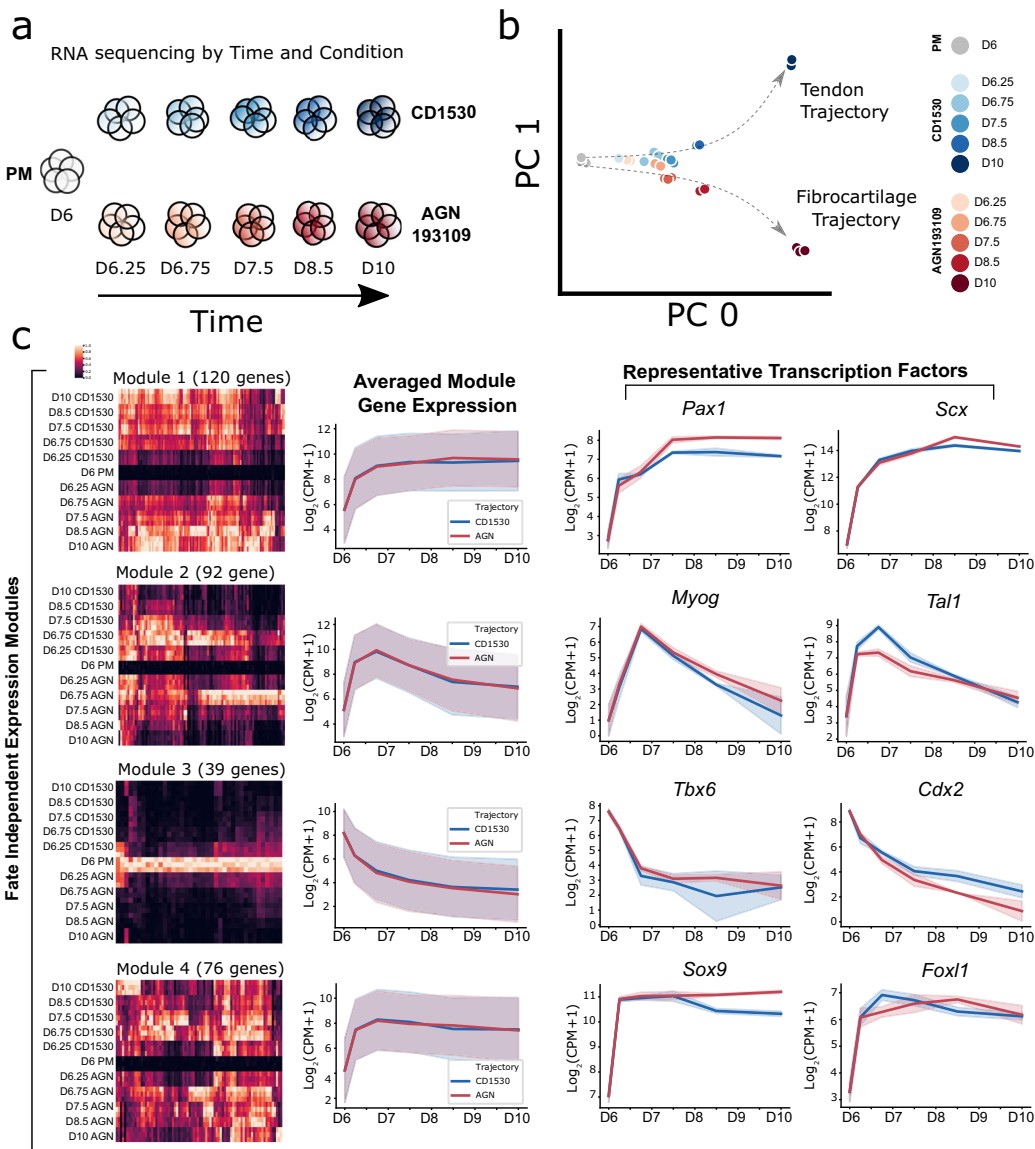

**Fig. 4 Transcriptional profiling and trajectory analysis by RNA sequencing reveal transcription factor modules associated with differentiation. a** Schematic for RNA-seq of CD1530 (blue) and AGN 193109 (red) temporal trajectories starting from the common starting point of D6 PM. Samples were collected at indicated timepoints and conditions for RNA-seq. **b** PCA of tendon and fibrocartilage trajectories. **c** Fate-independent gene modules represented as column-normalized heatmaps, averaged module gene expression line plots, and representative transcription factors. ($n = 3$ independent samples). Standard deviation is represented by shaded regions.

Expression patterns for the fibrocartilage trajectory modules (Modules 7 and 8) showed transiently expressed and steadily upregulated genes, respectively. Notably, the homeobox gene *Dlx5*, a regulator of chondrocyte differentiation and maturation[59–61], was found in Module 8 alongside well-established cartilage markers *Acan*, *Matn4*, and *Sox6* (Fig. 5, Supplementary Fig. 8, Supplementary Dataset 3). Although these genes also increased in the CD1530 condition compared to D6 PM, the levels at D10 were markedly lower compared to AGN 193109.

From trajectory module analyses, we identified several genes that have not been previously implicated in tendon or fibrocartilage development. To determine whether there may be in vivo relevance, we collected tails from E11.5-E14.5 embryos, spanning the stages of tendon and annulus fibrosus progenitor induction (E11.5 and E12.5) through differentiation (E14.5, Supplementary Fig. 9). Real-time qPCR analysis of the top genes contributing to PC0 showed dramatic upregulation at E14.5 compared to E11.5 or E12.5 stages for all genes (Supplementary Fig. 9).

**Extended culture of tendon and fibrocartilage cells leads to loss of fate and senescence.** To determine whether tendon and fibrocartilage phenotypes could be maintained under extended culture, we passaged cells at confluence (D10 and D14) and continued culturing in the presence of CD1530 or AGN 193109 (Supplementary Fig. 10). By D18, we found that while cells remained viable, they were no longer proliferative and ScxGFP expression was reduced in both media conditions. Analysis of phenotypic markers at D18 by qPCR showed a loss of tenogenic markers *Scx* and *Mkx* for the CD1530 condition. *Col2a1* was minimally expressed regardless of passage demonstrating no acquisition of a cartilage phenotype. In contrast, *Scx* expression modestly increased with extended 2D culture in the presence of AGN 193109, while *Tnmd* and *Col2a1* expression levels were unchanged (Supplementary Fig. 10). This suggests that while the

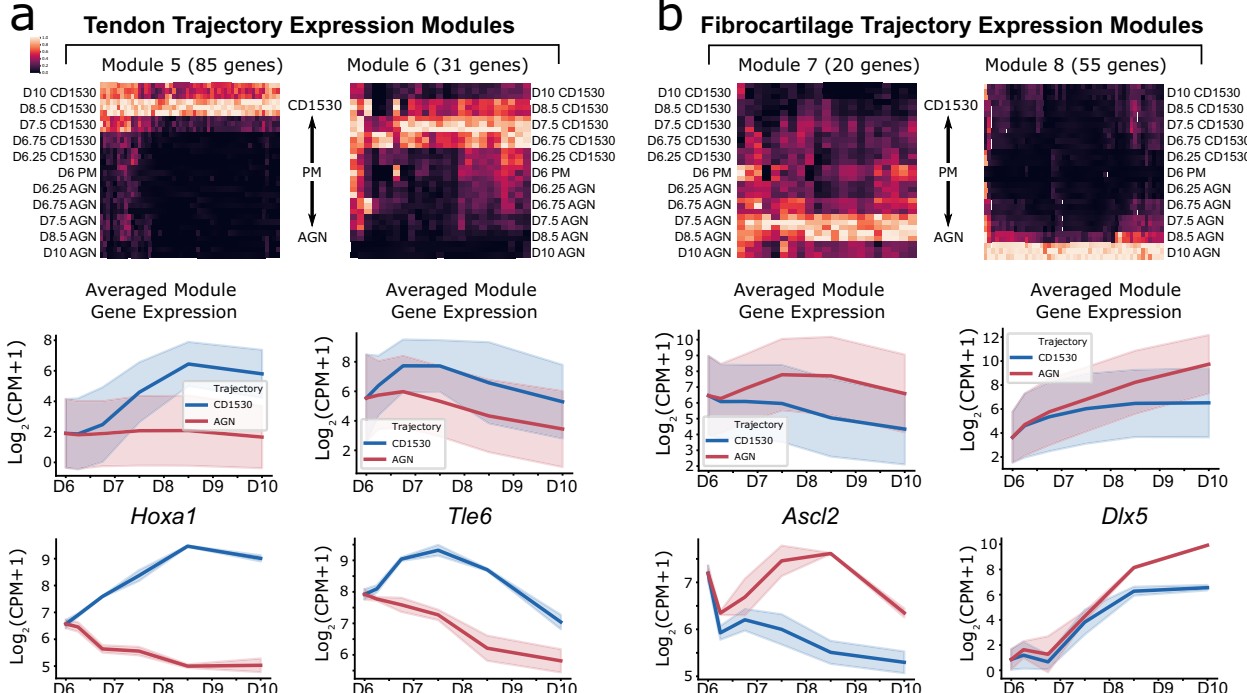

**Fig. 5 Transcriptional profiling and trajectory analysis by RNA sequencing reveals transcription factor modules associated with fate-specific tendon and fibrocartilage induction.** Fate-dependent gene modules for (**a**) tendon and (**b**) fibrocartilage trajectories represented as column-normalized heatmaps, averaged module gene expression line plots, and representative transcription factors. (*n* = 3 independent samples). Standard deviation is represented by shaded regions.

fibrocartilage AGN 193109-containing media was sufficient to maintain the phenotype of the cells, maintenance of the tenogenic phenotype may require additional cues.

**Differentiation of tendon and fibrocartilage cells under 3D uniaxial static tension.** Since tendon cells normally experience uniaxial mechanical tension during development, we next tested whether we could better sustain the tendon phenotype using a 3D tension culture system. mESCs were therefore differentiated toward D6 PM and then directly embedded into collagen gels held under static tension (Fig. 6a, b)[62]. We observed gradual cell-mediated contraction of the gels over a course of 11 days (D6-D17) with no difference in contraction between CD1530 and AGN 193109 conditions at any timepoint (Fig. 6b). To compare tendon and fibrocartilage induction in 3D culture relative to standard 2D culture, we harvested cells at D10 for qPCR analyses. In the CD1530 condition, *Scx* and *Acan* expression were similarly induced between 2D and 3D, while *Col1a1* increased in 3D gels (Supplementary Fig. 11). Gene expression for osteogenic, myogenic, and adipogenic markers did not show any differences between 2D or 3D culture, indicating no aberrant differentiation along these other lineages. With AGN 193109 treatment, we observed similar levels of tendon and cartilage markers between 2D and 3D conditions, but a surprising increase in the osteogenic marker *Osx* with 3D culture (Supplementary Fig. 11). Analysis of tendon-specific transcription factors identified from RNA-seq-derived modules showed upregulation of both *Hoxa1* and *Wt1* with 3D culture in CD1530. In contrast, no difference was observed for the fibrocartilage-specific transcription factors, *Alx4* and *Dlx5* with AGN 193109-treatment (Supplementary Fig. 11).

Since collagen is the main ECM component in both tendon and fibrocartilage tissues, we next assessed organized collagen deposition using second-harmonic generation (SHG) on whole-mount gels at D17. Acellular collagen gels were also imaged to distinguish the collagen gel biomaterial from cell-secreted collagen. Quantification of SHG images showed significant deposition of aligned collagen in the cell-seeded gels relative to acellular gels, with no detectable difference between CD1530 and AGN 193109 conditions (Fig. 6c). Whole-mount and multi-photon fluorescent imaging revealed the presence of ScxGFP+ longitudinally aligned cells in both conditions, but the appearance of rounded nodules only in the AGN 193109 condition (Fig. 6d, e). Staining of transverse sections showed that these rounded nodules stained intensely for type II collagen, indicating cartilage-like matrix (Fig. 7f). In contrast, cartilage nodules were not detected in any CD1530 constructs. Collectively, these data showed that 3D static tension supports tenogenic induction and differentiation, with the alignment of cells and deposition of collagen under CD1530 conditions. While ScxGFP cells also aligned in the AGN 193109 condition, the fibrocartilage phenotype was not appreciably altered under static tension based on gene expression and the presence of cartilaginous nodules, suggesting that biochemical cues are predominant for induction and maintenance of fibrocartilage.

**Dynamic tensile loading of 3D engineered tissues activate mechanotransduction pathways.** During embryogenesis, contractile muscle forces play a critical role in tendon and fibro-cartilage development[63–65]. To determine whether our engineered 3D tissues respond to mechanical loading, we adapted the STREX Cell Stretching System by designing 3D-printed molds to create custom PDMS dishes (Fig. 7a). D6 PM cell-seeded constructs were then generated and cultured within these dishes, which could be mounted onto the bioreactor system for dynamic tensile loading. Gels cultured within the custom dishes showed normal contraction and successful induction of ScxGFP (Fig. 7a–c). We validated the bioreactor for 3D gels by testing a range of bior-eactor inputs (2–15% stretch, 0.5 Hz) comparing elongation

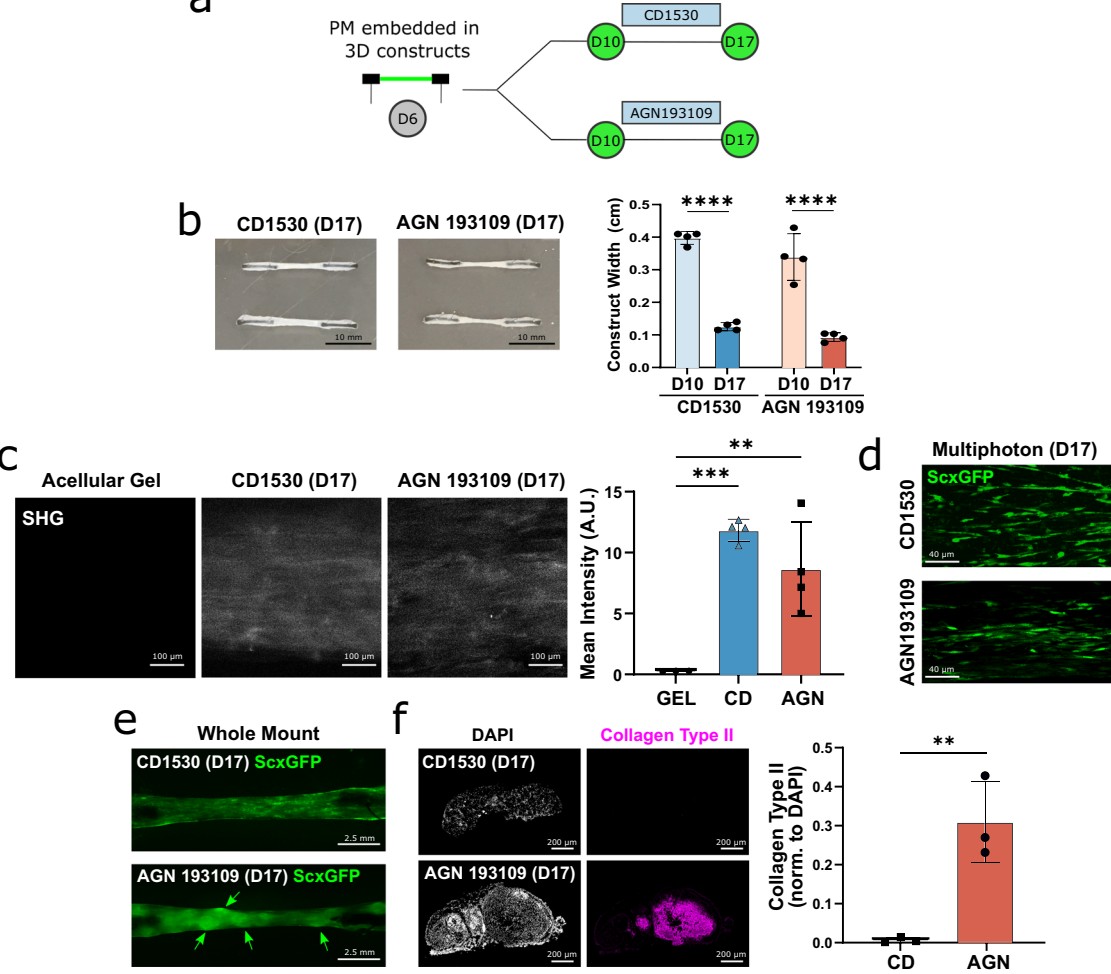

**Fig. 6 3D static tension aligns cells and collagen deposition in engineered tendon and fibrocartilage tissues. a** Schematic of experimental design. **b** Quantification of gel contraction over 17 days of culture in 3D ($n = 4$ independent gels, one-way ANOVA with Tukey's posthoc tests). Data represented as mean ± SD. **c** SHG imaging and quantification for aligned collagen deposition ($n = 4$ independent gels, one-way ANOVA with Tukey's posthoc tests). Data represented as mean ± SD. **d** Multiphoton imaging of aligned ScxGFP+ cells at D17 under 3D conditions. **e** Whole mount ScxGFP fluorescence imaging of 3D gels. **f** Immunostaining and quantification for collagen type II in transverse cryosections ($n = 3$ independent gels, unpaired two-sided Student's t-tests). Data represented as mean ± SD. **p < 0.01, ****p < 0.0001. Source data are provided as a source data file.

experienced by the gels against applied inputs. Image analysis showed a linear correlation of gel elongation with increasing applied stretch ($R^2 = 0.96$, Fig. 7d). Four hours of dynamic loading at 6% gel elongation (which is within the physiologic range for tendons in vivo[66]) resulted in upregulation of known mechanotransduction markers for both CD1530 (*Wwtr1*, *Mrtfa*) and AGN 193109 (*Yap1*, *Wwtr1*, *Trpv4*, *Mrtfa*) conditions, while 3% loading had no effect. These results establish a platform for studying tendon developmental mechanobiology.

## Discussion

In this study, we used an iterative approach informed by developmental cues and single-cell RNA sequencing to establish highly efficient methods for deriving tendon- and fibrocartilage-specific cells from mESCs through manipulation of the TGFβ, hedgehog, and retinoic acid signaling pathways. Using these models of tendon and fibrocartilage differentiation, we further define the trajectory of transcriptional signatures during differentiation and identify potential regulators distinguishing these phenotypes. Finally, we applied these media protocols to generate 3D engineered tendon and fibrocartilage tissues and established a system

for modeling tendon mechanobiology under static and dynamic culture conditions (Fig. 8).

To date, healing of dense, connective tissues (including tendon, ligament, meniscus, and the annulus fibrosus) remains poor in adults and treatment is largely limited to surgical repair[67–69]. Recently, a few studies showed that regenerative healing is possible during fetal and neonatal stages[70–74], however primary fetal and neonatal cells are not clinically viable options for transplantation. Although differentiated cells can be derived from pluripotent sources such as ESCs, from a translational perspective, there are ethical considerations for the use of these cells and practical challenges such as risk of teratoma formation[75,76]. These risks may be overcome by improving differentiation efficiencies or including selection markers[76]. In this study, we demonstrate that a directed differentiation approach efficiently generates tendon and fibrocartilage cells with transcriptional signatures comparable to in vivo embryonic cells, and can therefore be used to model developmental processes. While we refer to our nonchondrogenic *Scx*-expressing cell populations as tendon cells, it is likely that the 'tendon' phenotype here can be broadly applied to related tissues such as ligaments, which connect bone to bone. Although differences between tendons and ligaments emerge

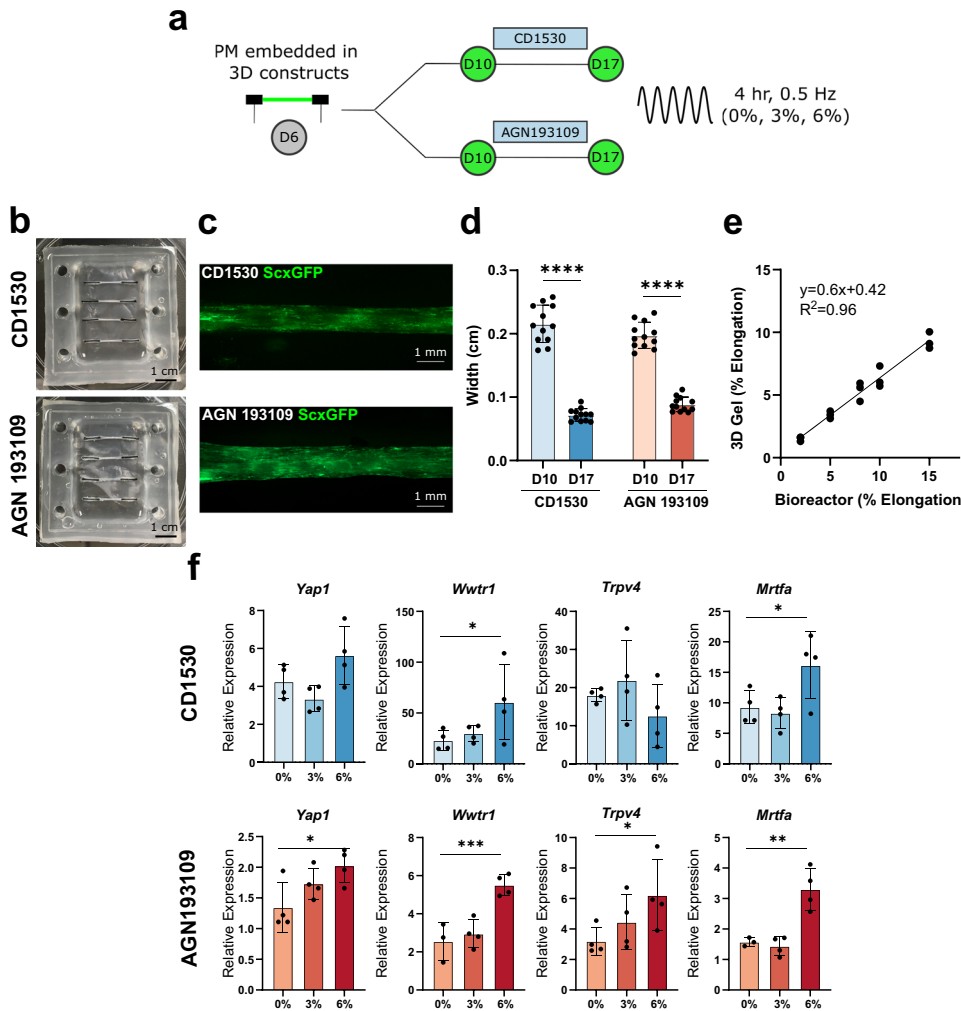

**Fig. 7 Dynamic tensile loading of 3D engineered tendon and fibrocartilage tissues activates mechanotransduction markers. a** Schematic of experimental design. Gels were dynamically loaded at D17. **b** Successful generation of contracted 3D cell-seeded gels in custom PDMS dishes at D17. **c** Whole-mount ScxGFP fluorescence imaging of 3D gels at D17. **d** Quantification of gel contraction ($n = 12$ independent gels, one-way ANOVA with Tukey's posthoc tests). Data represented as mean ± SD. **e** Validation of % elongation experienced by 3D gels with bioreactor loading ($n = 3$ independent gels, Pearson's correlation). **f** Real-time qPCR analysis of known mechanotransduction markers with 4 h of dynamic loading ($n = 4$ independent gels, one-way ANOVA with Fisher's LSD tests). Data represented as mean ± SD. *$p < 0.05$, **$p < 0.01$, ***$p < 0.001$, ****$p < 0.0001$. Source data are provided as a source data file.

postnatally with adaptation to local loading environments (and indeed, differences between anatomic tendons also emerge postnatally[77]), at these relatively early embryonic stages, there is no evidence that tendons and ligaments are transcriptionally distinct[23]. Ongoing research will focus on applying these protocols to human iPSCs, which may have more direct translational potential for tendon/ligament and fibrocartilage (meniscus/annulus fibrosus) repair.

Surprisingly, a pure cartilage population was not observed in our cultures despite the known role of TGFβ signaling in chondrogenesis[14,15], as cells expressing a chondrogenic signature also strongly expressed fibrous markers. These opposing responses to TGFβ may be due to the distinctive competence of paraxial mesoderm cells (vs. other progenitor cell types such as mesenchymal stem cells), since previous studies using mouse somitic cells also showed that TGFβ exerts a strong fibrogenic effect at the expense of chondrogenesis[78]. During limb development, TGFβ also induces bipotent *Scx+/Sox9+* progenitors that are subsequently allocated to either tendon (*Scx+* only) or cartilage (*Sox9+* only)[16]. This fate decision is mediated in part by

BMP signaling, which turns off *Scx* in a subset of progenitors while maintaining *Sox9*. Although BMP activation was shown to inhibit *Scx* expression in early limb development and BMP ligands are used to promote cartilage differentiation in vitro[17,28,37], manipulation of BMP signaling did not affect ScxGFP induction in our system when TGFβ signaling was also active. Our results suggest that while BMP4 signaling in the absence of TGFβ does indeed inhibit *Scx* expression, TGFβ may also induce the production of other members of the BMP family that further promote tenogenic induction (since supplementation with LDN-193189 resulted in a marked reduction of ScxGFP induction). Potential members may be BMPs 12, 13, and 14 (formerly known as GDFs 7, 6, and 5, respectively), which were shown to be involved in embryonic tendon development[79–82]. Recent studies also demonstrate a critical role for BMP signaling in tendon growth postnatally, indicating that BMP signaling is required for normal tendon development[83]. In addition, the sustained presence of TGFβ prevented the inhibitory effect of BMP4 signaling since only a modest decrease in ScxGFP cells was observed with no effect on tendon gene expression. This may be

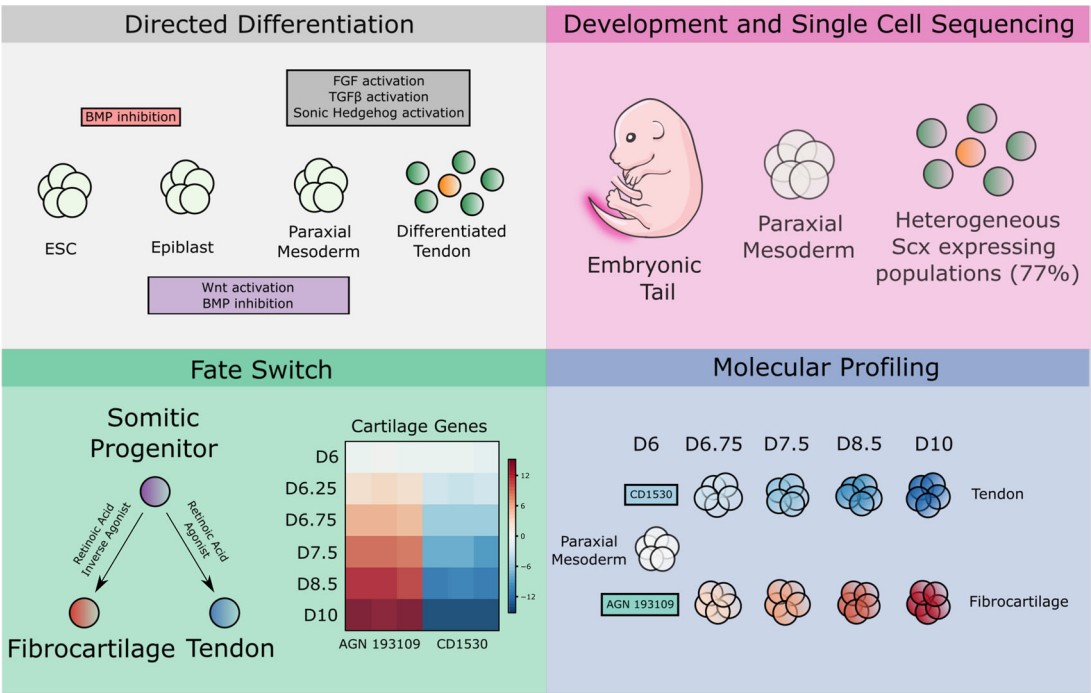

**Fig. 8 Conceptual schematic highlighting key findings.** Directed differentiation inspired by developmental events was used to induce ScxGFP tendon cells. Single-cell RNA-seq identified distinctive tendon and fibrocartilage cells with high transcriptional similarity to in vivo embryonic counterparts. Retinoic acid signaling was identified as fate-switch regulator between tendon and fibrocartilage cells. Molecular profiling identified regulators of tendon and fibrocartilage induction.

due to differences in timing/sequence of molecular factors or indicate that TGFβ exerts a more dominant or antagonistic role.

In these studies, we also identified retinoic acid signaling as a key driver of tendon vs fibrocartilage fates in the presence of TGFβ. Although retinoic acid signaling has been extensively studied in the context of skeletal and cartilage development, there are few studies in the context of tendon and fibrocartilage tissues. For tendon, retinoic acid molecules are implicated in tendon-muscle patterning in the eye and cranium[84,85]. Activation of retinoic acid signaling (in the absence of TGFβ) was also suggested to maintain stemness of tendon stem/progenitor cells while restricting differentiation along mesenchymal lineages, including tendon[86]. Of the three retinoic acid receptors (RARα, RARβ, and RARγ), RARγ is essential in the cartilage growth plate for skeletal development, but functions in combination with either RARα or RARβ since single null RARγ mutants are normal[87]. The RARs can act as transcriptional activators or repressors, depending on ligand binding. In the absence of ligand, RARγ functions as a repressor which results in chondrogenic activity. Repressor activity can be enhanced by using synthetic retinoids that act as inverse agonists, such as AGN 193109. Ligand binding with a standard agonist on the other hand (such as ATRA), results in repression of chondrogenesis[48]. Here, we applied synthetic retinoic acid molecules (AGN 193109 and CD1530) in the presence of TGFβ to differentiated paraxial mesoderm and showed that the anti-chondrogenic effects of RARγ agonism were independent of fibrous differentiation. We therefore propose that TGFβ drives the activation of fibrous markers while retinoic acid signaling toggles chondrogenesis. Unlike with BMP signaling, there does not seem to be an antagonistic relationship between TGFβ and retinoic acid pathways, although it has been suggested that the anti-chondrogenic activities of retinoic acid activation is through suppression of Smad2/3 phosphorylation[88]. Interestingly, recent research indicates that the tenogenic/fibrous functions of TGFβ signaling is likely through non-canonical (ie. non-Smad)

pathways, suggesting a mechanism by which TGFβ signaling can distinguish fibrous vs chondrogenic differentiation[83,89]. Interestingly, the activity of retinoic acid signaling appears highly context-dependent. During paraxial mesoderm specification, we found that early activation of retinoic acid signaling resulted in differentiation toward a neural fate. When paraxial mesoderm was successfully induced however, retinoic acid signaling functioned in concert with TGFβ to regulate tendon vs fibrocartilage. In our cultures, activation or inhibition of retinoic acid signaling at these latter stages did not induce a stem-like phenotype as was previously reported in adult-derived tendon stem/progenitor cells[86], since both CD1530 and AGN 193109 treatment groups showed robust upregulation of tenogenic differentiation markers. Further, sustained culture in 2D in the presence of CD1530 did not maintain proliferation potential. Other studies showed that within the context of BMP signaling, retinoic acid signaling does not inhibit chondrogenesis, but induces chondrocyte hypertrophy instead[90]. In limb bud cultures, retinoic acid signaling resulted in the suppression of chondrogenesis without impacting myogenesis[50]. Thus, the effect of retinoic acid activation or antagonism highly depends on other signaling pathways to provide contextualizing signals, as well as on the competence of the cell types experiencing the signaling. Future studies will further define interactions between TGFβ, BMP, and retinoic acid signaling pathways in tendon/fibrocartilage differentiation in vivo and in vitro and test additional molecular regulators of these fates.

Successful induction efficiencies of >90% enabled the use of RNA-seq to profile transcriptional trajectories defining tendon and fibrocartilage differentiation. We were particularly interested in early kinetics since the transcription factors that regulate tendon and fibrocartilage induction have yet to be identified. Transcriptional profiling of tendon and fibrocartilage fate trajectories revealed several transcription factors associated with induction, both fate-independent and -dependent. These included

several transcription factors that correlated with *Scx* expression pattern. Analysis of in vivo embryonic mouse tails showed upregulation of several of these molecules in concert with tendon and fibrocartilage differentiation. A subset of these were also identified in previous microarray and sequencing datasets of embryonic tendon cells[22,23], further demonstrating the potential of our differentiation models to recapitulate in vivo developmental processes. Future work will test the function of these transcription factors in tendon and fibrocartilage specification through gain or loss of function experiments. These findings provide new insight in the molecular regulation of these clinically challenging, dense connective tissues. Further, since definitive phenotypic markers remain limited for these tissues, our data now provide comprehensive transcriptional signatures associated with the acquisition of tendon and fibrocartilage cell fates.

Although the development and maintenance of fibrous vs cartilaginous tissues is typically thought to be driven by mechanical loading demands (tensile and compressive forces, respectively)[2,3,11], our studies also suggest that molecular factors alone can control the switch between tendon and fibrocartilage fates under the same 2D environment. However, prolonged culture under 2D conditions resulted in progressive loss of tenogenic phenotype, consistent with prior reports[91–93]. We overcome this limitation by inducing tenogenesis under 3D culture conditions and showed improved differentiation compared to 2D culture. Surprisingly, the fibrocartilage condition appeared largely insensitive to 3D environment since tendon and cartilage genes were relatively unchanged with 2D or 3D culture. Although compressive loading is thought to drive chondrogenesis, cartilaginous nodules were still observed under static tension, suggesting that biochemical cues can drive fibrocartilage formation independent of mechanical signals. This difference in mechanical dependency between tendon and fibrocartilage is intriguing and supported by some evidence in the developmental literature. For tendons, loss of muscles results in early and complete loss of forearm tendons in the mouse limb[63]. In contrast, the meniscus is formed and maintained when muscles are absent, but dynamic muscle contraction is required for the maintenance of matrix markers[94,95]. Using our system, we can dissect mechanotransduction mechanisms underlying tendon and fibrocartilage development and precisely control loading parameters, which is not feasible in vivo during mouse embryonic stages.

Here, we demonstrate the utility of these high-efficiency protocols for studying tendon and fibrocartilage development, mechanobiology, and signaling, however, they can also be applied for high-throughput small molecule screening to identify drug targets for clinical applications. Moreover, the wide availability of genetic tools in the mouse system also allows the use of mESCs derived from mutant lines for disease modeling or test loss of gene function in vitro in the context of tendon and fibrocartilage induction. Future studies will test these applications to advance scientific understanding and develop potential therapies for regenerative repair of these dense connective tissues.

## Methods

**Tenogenic differentiation of mESCs by activation of TGFβ and hedgehog signaling.** mESCs were derived from ScxGFP tendon reporter mice[9]. Briefly, mouse blastocysts (E3.5-3.75) from ScxGFP mice were individually plated onto irradiated primary mouse embryonic fibroblasts (MEFs, $1 \times 10^5$ cells/cm², R&D Systems Cat #PSC001) for 4–5 days in ES media (DMEM with 1 mM non-essential amino acids, 1 mM sodium pyruvate, 2 mM L-glutamine, 10 μM beta-mercaptoethanol, 15% ES-qualified fetal calf serum, 50 μg/mL penicillin/streptomycin, 1000 U/mL LIF). Outgrowths of the inner cell mass were then manually isolated and replated and passaged several times until characteristic ES colonies were observed[96]. A total of 13 lines were obtained and genotyped for ScxGFP, which showed 8 ScxGFP+ and 5 ScxGFP- mESC lines. Experiments were carried out using 3 ScxGFP+ lines and a ScxGFP- line was cultured and differentiated in parallel for each experiment to set flow cytometry gates for GFP.

Undifferentiated mESCs were maintained with fetal bovine serum (FBS) and 20 ng/μL leukemia inhibiting factor (LIF) on MEF feeders[97]. All experiments were conducted by passage 12 or below. To differentiate mESCs, single-cell suspensions were cultured in Serum-Free Differentiation (SFD) media for 2 days with 500 nM LDN-193189 for embryoid body formation[98]. At day 2, SFD with 500 nM LDN-193189 and 5 μM CHIR was used for 4 days. On day 6, tenogenesis was induced using SFD with 10 ng/mL TGFβ1, 100 nM SAG, and 10 ng/mL FGF2. ScxGFP induction was determined using flow cytometry and detection of ScxGFPhi and ScxGFPlo cells was performed based on TGFβ1+ conditions as described below.

**Optimization of tendon differentiation by manipulation of retinoic acid signaling.** To enhance paraxial mesoderm differentiation, embryoid body formation was carried out as described above, followed by 4 days of SFD with 500 nM LDN-193189, 5 μM CHIR, and either 100 nM AGN 193109 or 30 nM ATRA. Tendon-specific differentiation was subsequently induced using 4 days of SFD with 10 ng/mL TGFβ1, 100 nM SAG, 10 ng/mL FGF2, and 1 μM CD1530. Fibrocartilage-specific differentiation was induced for 4 days in SFD with 10 ng/mL TGFβ1, 100 nM SAG, 10 ng/mL FGF2, and 100 nM AGN 193109.

**Flow cytometry.** Single-cell suspensions of dissociated cells were obtained at day 10 of differentiation and stained 1:1000 with DAPI (ThermoFisher) in 2% FBS in PBS. Flow cytometry was carried out on an LSRIIA instrument (BD Sciences) using FACSDiva v8.0.3. Flow cytometry analysis was carried out using FCS Express v7. Gating for ScxGFP positivity was performed with a negative control mESC line (derived from littermates) that did not contain the ScxGFP transgene and represented as a percentage of the parent gate. Flow cytometry gating strategy for ScxGFP is shown in Supplementary Fig. 12. ScxGFP high and low gates were formed by identifying local minima in TGFβ1 containing conditions.

**RNA isolation, reverse transcription, and real-time qPCR.** RNA was isolated by Trizol/chloroform (Invitrogen) extraction, followed by cDNA synthesis (SuperScript VILO, Invitrogen), and gene expression assessed by qRT-PCR (SYBR, Applied Biosystems). Data were calculated using $2^{-\Delta\Delta Ct}$ with *Gapdh*. All primer sequences can be found in Supplementary Table 1.

**Single-cell RNA sequencing.** Single-cell suspensions from cultured cells and E14.5 embryonic mouse tail cells were prepared and dead cell depletion performed (EasySep Dead Cell Removal (Annexin V) Kit, STEMCELL. Live cells were input into a 10x Genomics Chromium device. Single-cell RNA libraries were prepared using the Chromium Single Cell 3′ v3 Reagent Kit (10x Genomics) user guide. Sample libraries were pooled together at equimolar ratios and sequenced on an Illumina NextSeq 550 using 75-bp paired-end reads (high-output 75 cycle kit, cat no. 20024906), aiming for a sequencing depth of 20,000 reads/cell.

All subsequent data processing was conducted using the *cellranger* suite (version 3.1.0, 10X Genomics). Obtained FASTQ reads were aligned to the genome reference and mapped per cell using *cellranger* count. Multi-mapping reads were ignored during quantification. Gene-cell matrices were extracted and read into R and further analyzed using Seurat (version 3.0).

To exclude low-quality cells, the following quality control filters were applied and cells removed: cells with >7.5% of transcripts mapping to the mitochondrial genome, cells in the bottom 2% quantile (low quality) and top 2% quantiles (likely doublets/multiplets) of the transcripts detected per cell. Data of the resulting cells was normalized via the SCTransform method.

To confirm the identity of cultured cells, we used Seurat's integrated projection method, setting the scRNA-seq data from the embryonic tail as a reference and the cultured cells as the query. To account for batch effect differences, we used the Seurat alignment method to integrate the datasets. Developer defaults were used to conduct integration. The alignment leverages canonical correlation analysis to find linear relationships between features between datasets. Using these linear relationships, the method identifies a shared correlation space where accurate comparisons between datasets can be made. For each dataset, highly variable features were identified by taking the union of the top 3000 genes with the highest dispersion across both datasets. Canonical correlation analysis was applied upon these highly variable features to identify common sources of variation between the datasets. The first 30 correlational components were used to align the datasets and the resulting dataset was used for further analysis.

Principle component analysis was performed on the batch-corrected dataset. After visual inspection of the resulting components and contributions via screeplot, 30 dimensions were used for unsupervised clustering and 2D projection of the data. Unsupervised clustering was done using the smart local moving algorithm () at a resolution of 0.2 to identify the major clusters. 2D projection and visualization of the clustering results were done using the uniform manifold approximation and projection (UMAP) method. UMAPs were generated using the same number of dimensions as with the clustering. Other gene expression plots were generated using Seurat base functions and ggplot2 v3.3.0. Gene ontology was performed using enrichR (2016 version)[99].

**RNA sequencing.** Time course RNA-seq was performed by preparing three biological replicates of samples from the undifferentiated PM and samples from the

tendon and fibrocartilage differentiation protocols at days 0.25 (6 h), 0.75 (18 h), 1.5, 2.5, and 4. RNA extraction was performed using Qiagen's RNeasy Plus Universal mini kit according to the manufacturer's instructions. Libraries were prepared using the NEB Next Ultra NRA Library Prep Kit for Illumina Sequencing in accordance with the manufacturer's instructions. The sequencing libraries were validated on the Agilent TapeStation (Agilent Technologies), and quantified by using Qubit 2.0 Fluorometer (Invitrogen) as well as by quantitative PCR (KAPA Biosystems).

The sequencing libraries were clustered on 1 lane of a flowcell. After clustering, the flowcell was loaded on the Illumina 4000 HiSeq instrument according to the manufacturer's instructions. The samples were sequenced using a 2 × 150 bp Paired End (PE) configuration. Image analysis and base calling were conducted by the HiSeq Control Software v3.6.3. Raw sequence data (.bcl files) generated from Illumina HiSeq was converted into fastq files and de-multiplexedusing Illumina's bcl2fastq v2.20 software. One mismatch was allowed for index sequence identification.

Spliced transcripts were aligned and mapped to the Mus musculus reference genome (GRCm38) using STAR Aligner v2.7[100]. Feature counting was subsequently performed with featureCounts v1.6.0[101]. Normalization, estimation of dispersion effects, and differential gene expression (DGE) were performed in R software v3.6.3 using edgeR v3.28.1[102]. Hierarchical clustering was implemented in SciPy v1.3.0[103]. All images were visualized in Python v3.7.3 with Seaborn v0.9.0 and Matplotlib v3.1.0[104].

**Proteomic analysis of receptor tyrosine kinases**. The Mouse Phospho-Receptor Tyrosine Kinase (RTK) Array Kit (Catalog # ARY014, R&D Systems) was used to detect phosphorylation levels of 39 RTKs in day 10 differentiated cells in accordance with the manufacturer's protocol. Two hundred micrograms of total protein lysate from each sample was used per membrane. Protein concentration was determined using Pierce BCA Protein Assay Reagent (ThermoFisher Scientific). Following processing, each array membrane was exposed to X-ray film for 1 h and developed using a JPI X-ray film processor. Relative density of positive array hits was analyzed using ImageJ v1.53e software gel analysis and background subtraction applied using the included negative PBS control.

**Collection of mouse embryos**. Mouse embryos were collected from ScxGFP dams based on the time of conception according to Institutional Animal Care and Use Committee at the Icahn School of Medicine at Mount Sinai approved procedures (Protocol #IACUC-2014-0031) and consistent with animal care guidelines. Whole-mount imaging of embryos was carried out immediately at the time of harvest using Leica stereoscope fitted with GFP filter. Tails were then immediately placed in Trizol for RNA isolation.

**Generation of 3D engineered tissues and dynamic loading**. mESCs were differentiated toward D6 PM under 2D conditions, trypsinized, and embedded into 3D type I collagen gels at 250,000 cells per 2 mg/mL PureCol[62]. Cell-seeded gels were cast into sterilely prepared PDMS plates between two prolene sutures pinned in place by minutien pins. After casting, gels were placed in a humidified 37 C incubator for 1 h to allow the gel to set before flooding with SDF media containing 10 ng/mL FGF2, 10 ng/mL TGFβ1, and 100 nMSAG. Tendon-specific conditions also included 1 μM CD1530 while fibrocartilage-specific conditions included 100 nM AGN 193109. For dynamic loading experiments, a custom 3D-printed mold was used to cast PDMS dishes that fit on the STREX Cell Stretching System. Cell-seeded gels were then created in these custom dishes as described above and cultured for 7 days in CD1530- or AGN 193109-containing media until full contraction. Validation of loading parameters was carried out by imaging gels during varying bioreactor inputs (2–15% elongation). The elongation experienced by gels was measured for each input using ImageJ. A calibration curve was then generated. Based on the literature indicating physiologic tendon loading of 6% elongation, we tested 3% and 6% elongation at 0.5 Hz for 4 h. Cell-seeded gels were collected immediately after loading for qPCR. Imaging of gels prior and after loading did not show changes in overall morphology or visible indications of damage.

**Second-harmonic generation two-photon imaging**. Imaging was performed on whole-mount constructs in Opti-MEM medium without phenol red using the Olympus FVMPE-RS multiphoton laser scanning microscope with a water immersive objective at ×25. Acellular gels were imaged in parallel and used as background controls. Multiphoton excitation was performed with Dual Line Insight X3 laser at an excitation of 880 nm. Second-harmonic generation (SHG) and ScxGFP were recorded by GaAsP PMT detectors (4 G and 3 G, respectively). All parameters (ie, laser intensity, gain, high voltage, and offset) were selected to minimize background noise without oversaturation and were held constant for all constructs. Images were converted using the Bio-Formats plugin and batch-processed in ImageJ. Mean pixel intensity were analyzed to quantify SHG signal and collagen density.

**Immunofluorescence analyses**. Engineered tendon constructs were fixed in 4% paraformaldehyde for 30 min, infiltrated in 15% and 30% sucrose (24 h), and embedded in OCT medium. Transverse cryosections (12 μm) were collected and

immunostaining for collagen type II carried out using anti-COL2α1 (II-II6B3, 1:100, Developmental Studies Hybridoma Bank) with secondary detection by strepdavidin-Cy3 (016-160-084, 1:200, Jackson ImmunoResearch) with the M.O. M. kit (Vector Labs). Fluorescence imaging was carried out using the Zeiss Axioimager with Apotome optical sectioning and quantification performed in ImageJ.

**Statistical analyses**. Quantitative results are presented as mean ± standard deviation (SD). ANOVA was used for multiple comparisons; where significance was detected, posthoc testing was then carried out (Graphpad Prism). All other quantitative analyses were analyzed using unpaired Students t-tests. Statistical analyses are indicated in figure legends. Significant outliers were detected using Grubb's test (Graphpad Prism). Significance was determined at $p < 0.05$.

**Reporting summary**. Further information on research design is available in the Nature Research Reporting Summary linked to this article.

## Data availability

The scRNA-seq and RNA-seq data generated in this study have been deposited in the GEO repository database under accession codes "GSE154525" and "GSE154397". All other quantitative datasets including qPCR analyses, protein analyses, and immunostaining quantification are available in Supplementary Information/Source Data file provided with this paper. Source data are provided with this paper.

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

## Acknowledgements

This work was supported by funding from the National Institute of Arthritis and Musculoskeletal and Skin Diseases (NIAMS) to A.H. (R01 AR069537) and D.K. (F31 AR073626), and from NYSTEM to A.H. (C32570GG). We also acknowledge Dr. Ronen Schweitzer for providing ScxGFP mice, the Mouse Genetics and Gene Targeting Core for the derivation of mESCs from ScxGFP mice, the Flow Cytometry Core, qPCR Core, and the Human Immune Monitoring Center for single-cell RNA sequencing. The monoclonal antibody II-II6B3 developed by T.F. Linsenmayer was obtained from the Developmental Studies Hybridoma Bank, created by the NICHD of the NIH and maintained at the University of Iowa, Department of Biology, Iowa City, IA 52242.

## Author contributions

D.K. and A.H. conceived and designed the experiments. D.K. and A.M. performed tissue culture experiments and analyses. D.K. and R.P. performed bioinformatic analyses of the single-cell and bulk RNA sequencing datasets. The manuscript was written by D.K. and A.H. and edited and approved by all authors.

## Competing interests

The authors have no competing interests to declare.
