## [Peer Review File · Nature Communications]

Reviewers' Comments:

Reviewer #1:

Remarks to the Author:

A fundamental question in musculoskeletal biology is how the embryonic stem cell differentiate into tenocytes and fibrocartilage cells, and how these cells form tendon structure and fibrocartilage. The answer to the questions is essential for understanding tendon and fibrocartilage injury mechanisms and also for developing better treatment strategies. In this study, the authors used Scx-GFP mESC differentiation model in vitro to simulate the development of embryonic stem cells differentiation into tendon and fibrocartilage cells. The authors compared the similarity between the in vitro derived tenocytes and E14.5 embryonic tail tendon cells and confirmed this simple in vitro differentiation model to be suitable to study stem cell differentiation to tendon and fibrocartilage lineages. This work, to our knowledge, represents the first efforts to profile key transcription switches in developmental stages.

This study has yielded a trajectory of transcriptional signature of tendon development within embryonic differentiation stage. However, it did not identify any new pivotal genes that regulate tendon and fibrocartilage development. The roles of TGF-beta and Hedgehog signaling pathways in tendon development have been well established, and retinoic acid signaling pathway has also been well investigated in embryonic development. In fact, the transcriptional profiling in this study does not offer any exciting new information in these regards.

Moreover, what is the ultimate goal of this paper since the authors have been able to generate tendon and fibrocartilage cells with transcriptional signatures comparable to in vivo cells (line 315)? What would be the potential future use of these cells? Most of the conclusions are based on computer modeling with single cell analysis. There is no 'wet lab' evidence here.

For musculoskeletal tissues, an important factor affecting the tendon and fibrocartilage development is biomechanical loading. For example, bone formation in vitro model generally requires biomechanical loading like shear stress to form the bone like tissue, in addition to induction factors for cell differentiation. Therefore, this study may include biomechanical loading to mESC in vitro model for more accurate simulation of in vivo environment of tissue formation.

Also, ScxGFP-Hi and ScxGFP-lo seem to be separate populations. Did the authors test where these separate populations may be localized within developing embryos/prenatal/fetal mice? Can the authors say whether ScxGFP-Hi and ScxGFP-lo represent separate populations of musculoskeletal structures, such as tendon vs ligament?

Specific Comments:

1. It is important to describe the timing of the addition of certain factors to cultures. In line 108, 'To improve tenogenesis...' the authors explored the use of smoothened agonist. What was the time point when this was added during the differentiation scheme in Fig. 1? Same goes for addition of SB-43154 in line 102. in lines 106 to 108: The authors should also state the specific time point of 50% induction efficiency.
2. The authors need to clarify the time point when cultures were assessed with specific experiments. For example, in Fig. 1 the exact time points of flow cytometry assessment of the cultures and performance of real time qPCR experiment in G should be specified.
3. In many cases, the poor resolution of the images makes it difficult to make out which genes were analyzed in 2D. These genes are separated into tendon, proliferating, and cartilage genes. Did the authors test for the presence of genes for bone, ligament, muscle, adipose markers or any other musculoskeletal genes?
4. Lines 329-331: If manipulating BMP signaling did not affect ScxGFP induction within this model, how does that affect the cells the authors are analyzing? The results seem questionable if these experiments cannot in some way replicate well known tendon research from other papers.

5. It is unclear how the authors are harvesting mESCs from transgenic ScxGFP mice. Scleraxis in developing mice is a marker for all connective tissue including tendons as well as ligaments. The authors isolated ScxGFP mESCs (which is not described) and then used the addition of various factors to initiate tenogenic differentiation. How do the authors control for differentiation into other cell lines? What other markers did they test to make sure that their cultures are relatively 'pure' tendon progenitors and not expressing markers for other connective tissues like ligament? In vitro differentiation can result in unintended differentiation into other lines. What sort of quality control did these cultures undergo to make sure that cultures were relatively 'pure'?

6. Fig. 1G. Authors may include the controls of undifferentiated mESC. This is important for understanding the level changes during differentiation.

7. Fig. 4. The specific markers used in Fig. 3 should be repeated in Fig. 4.

8. Fig. 3I. The gene expression is only investigated at a single time point at the final stage. More time points may be tested for obtaining the gene changes so that differentiation process can be better defined.

Reviewer #2:

Remarks to the Author:

In consideration of the manuscript at the current state, I do not recommend that this paper be published on this journal.

Using an iterative approach informed by developmental cues and scRNA-Seq, the authors establish directed differentiation models to generate tendon and fibrocartilage cells from mESCs by activation of TGF β and hedgehog pathways, achieving a high induction efficiency. They also further define the trajectory of transcriptional signatures during differentiation and identify potential regulators distinguishing these phenotypes.

However, I consider the overall innovation of this study slightly insufficient. The authors apply widely used transcription factors to induce ESCs that are few studied in this research area and the results do fill in the blanks. While ESCs also have limitation in source of acquisition and transplantation for regeneration and repair in vivo (Lui PP, Stem Cells Cloning. 2015; 8:163-174. / Shojaee A, et al. Stem Cell Res Ther. 2019; 10(1):181.) compared to other previous researches using other kinds of stem cells as mentioned in the manuscript.

Besides, the authors culture these cells under 2-dimensional environment and generate tendon and fibrocartilage cells with transcriptional signatures comparable to in vivo embryonic cells. As many previous studies use 3-dimensional culture to induce tendon cells (Chen X, et al. Sci Rep. 2012; 2:977. / Theodossiou SK, Schiele NR. BMC Biomed Eng. 2019; 1:10.1186.) and some indicate 3-dimensional culture may be more suitable for induction and closer to the environment in vivo (Barsby T, et al. Tissue Eng Part A. 2014; 20(19-20):2604-2613. / Bavin EP, et al. Front Vet Sci. 2015;2:55.). Based on the above two points, I consider this model has certain limitation in wide application.

Also, it would be more convincing and comprehensive if the authors add some other experiment results in vivo and vitro (such as some morphological results, proteomics experiments, repairation results and so on. Komura S, et al. Sci Rep. 2020; 10(1):3992. / Tan GK, et al. Elife. 2020; 9: e52695.)

In addition, some figures' resolution in the paper is too low for readers to figure out the words (fig. 2-4) and these figures could also use some aesthetical polishing.

Reviewer #3:

Remarks to the Author:

In "Transcriptional profiling of mESC-derived tendon and fibrocartilage cell fate switch," Kaji and co-authors establish what the reviewer considers to be a highly transformative, developmentally-based approach to generating tendon and fibrocartilage cells from mouse embryonic stem cells (mESCs). The approach involves choreographed activation of the TGF β and hedgehog pathways

and has remarkable efficiency. The serum-free approach to generating these cells stands to transform tissue engineering. The molecular profiling of tendon and fibrocartilage differentiation trajectories by RNA sequencing is exquisite.

The reviewer is highly enthusiastic about the contribution, and has only praise for the study and a few questions that the authors could, if they choose, consider adding to the (already dense) discussion. The questions relate to kinetics and mechanobiology.

First, concerning the choreography of the TGF β and hedgehog pathways, how does the critical retinoic acid timing stay on schedule *in vivo*, with a more complicated circulation of factors such as TGF β ? Do the authors have any insight into how these detailed and apparently kinetics are activated in parallel for successful differentiation in development?

The second, related question relates to mechanical cues. TGF β is a highly mechanosensitive protein. Do the authors have any insight into how the cells, especially the fibrocartilaginous cells, are able to ignore their mechanical environments to reach the desired endpoints? The groups of Stavros Thomopoulos, Helen Lu, Johnna Temenoff, and many others who are interested in growing both tendon and the fibrocartilaginous tendon-to-bone insertion put much stock in the need for mechanical loading to drive cells to the desired phenotype in tissue engineered constructs, as do those of us who model the problem. Are all of these mechanical treatments simply ways of mechanically driving the TGF β and hedgehog choreography that the authors have characterized? The reviewer does not expect the authors to have definitive answers to these questions, but if they have insight the reviewer would value their thoughts on these matters in the discussion.

The final question relates to the growth plate. Given the remarkable switching that the authors have found, how do cells in the tendon-to-bone insertion choose and maintain their phenotype? With fibrocartilaginous and tendinous cells almost abutting one another, how do they shield one another? Modelers of the problem assert that mechanics locks in the fate niche, but the authors appear to have overturned this by showing that the fate can be determined independent of mechanics. How does it work instead?

Trivialities:

The manuscript is very well written and the figures are beautiful. Very minor suggestions to consider:

(1) In figure 4, the reviewer would have rather seen identical ordinate axes on all graphs. If there is a reason not to do this, a word on what is gained from having the different ordinate axes in the discussion would be appreciated as the reviewer was not sufficiently refined to pick it up on his own.

(2) Figure 4 has so much interesting information that it was in places unreadable. Because this journal allows more than four figures, would it make sense to split the figure into two separate figures that span two journal pages?

Congratulations on an outstanding contribution.

Guy Genin, Washington University

Reviewer Comments

Reviewer 1:

A fundamental question in musculoskeletal biology is how the embryonic stem cell differentiate into tenocytes and fibrocartilage cells, and how these cells form tendon structure and fibrocartilage. The answer to the questions is essential for understanding tendon and fibrocartilage injury mechanisms and also for developing better treatment strategies. In this study, the authors used Scx-GFP mESC differentiation model in vitro to simulate the development of embryonic stem cells differentiation into tendon and fibrocartilage cells. The authors compared the similarity between the in vitro derived tenocytes and E14.5 embryonic tail tendon cells and confirmed this simple in vitro differentiation model to be suitable to study stem cell differentiation to tendon and fibrocartilage lineages. This work, to our knowledge, represents the first efforts to profile key transcription switches in developmental stages.

Reviewer 2:

In consideration of the manuscript at the current state, I do not recommend that this paper be published on this journal.

Using an iterative approach informed by developmental cues and scRNA-Seq, the authors establish directed differentiation models to generate tendon and fibrocartilage cells from mESCs by activation of TGF β and hedgehog pathways, achieving a high induction efficiency. They also further define the trajectory of transcriptional signatures during differentiation and identify potential regulators distinguishing these phenotypes.

Reviewer 3:

In "Transcriptional profiling of mESC-derived tendon and fibrocartilage cell fate switch," Kaji and co-authors establish what the reviewer considers to be a highly transformative, developmentally-based approach to generating tendon and fibrocartilage cells from mouse embryonic stem cells (mESCs). The approach involves choreographed activation of the TGF β and hedgehog pathways and has remarkable efficiency. The serum-free approach to generating these cells stands to transform tissue engineering. The molecular profiling of tendon and fibrocartilage differentiation trajectories by RNA sequencing is exquisite.

The reviewer is highly enthusiastic about the contribution, and has only praise for the study and a few questions that the authors could, if they choose, consider adding to the (already dense) discussion. The questions relate to kinetics and mechanobiology.

Author Response: Thanks to all of the reviewers for their careful and thoughtful reading of the manuscript and suggestions for improvement. Based on reviewer recommendations, we have substantially improved the manuscript by adding several new experiments suggested, including BMP experiments, extended 2D culture, 3D culture as engineered tissues (with static and dynamic tensile loading), proteomics, second harmonic generation imaging analysis of collagen, *in vivo* validation of novel markers identified *in vitro*, and additional analyses of the transcriptional data. All comments and edits have been addressed (see below table). Changes to the text addressing reviewer comments and new data is indicated by highlighting in the main manuscript. The manuscript now includes 8 main figures, 1 table, 9 supplementary figures, and 5 supplementary tables.

Response to Reviewer 1 Comments

1. Moreover, what is the ultimate goal of this paper since the authors have been able to generate tendon and fibrocartilage cells with transcriptional signatures comparable to in vivo cells (line 315)? What would be the potential future use of these cells? Most of the conclusions are based on computer modeling with single cell analysis. There is no 'wet lab' evidence here.

We believe these differentiation platforms have very useful applications in high-throughput small molecule screening (to identify inducers and inhibitors for tendon and fibrocartilage), for modeling development, or for modeling disease (Irion, Nostro, Kattman, & Keller, 2008). For example, we can derive mESCs from mouse mutants or use siRNA knockdown to study cell-fate acquisition or switches. This is also a useful platform to study tendon mechanobiology, since tensional forces are a critical determinant of tendon development. We test this mechanobiology application in the current study and now include new wet-lab experiments that successfully create 3D engineered tendons and fibrocartilage tissues with these protocols and apply bioreactor loading to activate selected mechanotransduction molecules. Since muscle forces are difficult to control in the embryo, we believe this is a highly significant application.

Revision:

- New data shown in Figure 6, Figure 7, and Supplementary Figure 9
 - New results described in Pg 15-17
 - Additional discussion in Pg 21-22
2. This study has yielded a trajectory of transcriptional signature of tendon development within embryonic differentiation stage. However, it did not identify any new pivotal genes that regulate tendon and fibrocartilage development. The roles of TGF-beta and Hedgehog signaling pathways in tendon development have been well established, and retinoic acid signaling pathway has also been well investigated in embryonic development. In fact, the transcriptional profiling in this study does not offer any exciting new information in these regards.

A review of the existing literature shows that the role of retinoic acid (RA) in the musculoskeletal system has been primarily studied in the context of cartilage and skeletal development. In the context of tendon development, two papers suggest a role for RA signaling in tendon-muscle patterning of the eye and cranium (Comai et al., 2020; McGurk, Swartz, Chen, Galloway, & Eberhart, 2017). In the context of fibrocartilage, pubmed searches for "retinoic acid fibrocartilage" "retinoic acid meniscus" and retinoic acid "annulus fibrosus" returned no relevant results. Therefore, to our knowledge, the role of RA in driving cell fate decisions between tendon and fibrocartilage has not been studied or reported.

One exciting key finding from our study is the ability to regulate chondrogenesis independent of fibrogenesis – we were able to completely preserve the fibrogenic/tenogenic program while activating or repressing the chondrogenic program in parallel. This intriguing finding is likely due to the distinctive functions of TGF-beta and RA signaling.

Furthermore, our studies show an interesting modifying effect of hedgehog signaling on cell responsiveness to TGFbeta signaling. To our knowledge this has also not been previously reported *in vivo* or *in vitro*.

Our RNA-seq profiling of tendon and fibrocartilage trajectories also revealed several new genes that have not been previously associated with these tissues. We now show that several of these genes identified from our transcriptional analyses are upregulated with tendon/fibrocartilage development in the embryonic tail from E11.5/E12.5 induction to E14.5 differentiation stages, suggesting *in vivo* relevance. We also use selected markers (Hoxa1, Wt1, Alx4, and Dlx5) as fate-specific markers and test them in 3D culture studies. Direct testing of gene function *in vivo* is outside the scope of the current study which is already quite dense, however it is a promising avenue for future loss or gain of function studies.

Revision:

- New data shown in Supplementary Figure 7, Supplementary Figure 9
 - New results described in Pg 14, 16
 - Additional discussion in Pg 19-21
3. For musculoskeletal tissues, an important factor affecting the tendon and fibrocartilage development is biomechanical loading. For example, bone formation in vitro model generally requires biomechanical loading like shear stress to form the bone like tissue, in addition to induction factors for cell differentiation. Therefore, this study may include biomechanical loading to mESC in vitro model for more accurate simulation of in vivo environment of tissue formation.

We agree this is an important aspect of tendon/fibrocartilage development and now include experiments generating 3D engineered tissues to study differentiation within a tensioned collagen gel. We show that cells undergo successful phenotypic induction with preservation of phenotype over time and the 3D environment induces alignment of cells and collagen. We further adapt and validate a bioreactor system to apply dynamic loading to these engineered tissues and show upregulation of mechanotransduction markers.

Revision:

- New data shown in Figure 6, Figure 7, and Supplementary Figure 9
 - New results described in Pg 15-17
 - Additional discussion in Pg 21-22
4. Also, ScxGFP-Hi and ScxGFP-lo seem to be separate populations. Did the authors test where these separate populations may be localized within developing embryos/prenatal/fetal mice? Can the authors say whether ScxGFP-Hi and ScxGFP-lo represent separate populations of musculoskeletal structures, such as tendon vs ligament?

Our scRNA-seq results did not suggest distinctive signatures associated with ScxGFP-hi or lo. In our experience, tendons and ligaments in the embryo express comparable levels of ScxGFP. We also consulted with Dr. Ronen Schweitzer who created this mouse and he has also not identified any differences in ScxGFP expression between these tissues. We believe that the cells

generated in this study are likely applicable for ligaments as well and now include consideration of tendon vs ligament in the Discussion. It is possible that other connective tissue subtypes such as muscle connective tissue or fascia are more similar to ScxGFP-lo cells (lineage tracing with ScxCre show that these other connective tissues also come from Scx-lineage cells). Indeed, recent scRNA-seq data from Dr. Fabien LeGrand’s lab report that muscle connective tissue cells appear to express known tendon markers, such as Scx and Tnmd (Giordani et al., 2019). To address potential differences in Scx-expressing cells across these tissues, we carried out comparisons of our dataset to existing datasets for lung, muscle, and kidney from Tabula Muris (Figure 1A below), and found that Scx-expressing cells display rather wide heterogeneity depending on tissue type. Comparing our transcriptional signatures to that of these other non-tendon Scx populations show that our populations are most like the limb muscle Scx population and cluster 13 of the lung Scx population, but not Scx populations in the kidney or cluster 5 of the lung (Figure 1B). We include these results below for the reviewer’s benefit but have not included in the manuscript.

Figure 1: scRNA-seq comparison of tendon Scx transcriptional signature (cluster 0) with transcriptional signatures reported from non-tendon tissues. (A) Definitive Scx populations were only identified for kidney (cluster 9), limb muscle (cluster 3), and lung (clusters 5 and 13). (B) Heatmaps show correlation of these Scx populations with our cluster 0 Scx dataset (more similarity is indicated by greater amount of red signal).

Revision:

- Additional discussion in Pg 18

5. It is important to describe the timing of the addition of certain factors to cultures. In line 108, ‘To improve tenogenesis...’ the authors explored the use of smoothed agonist. What was the time point when this was added during the differentiation scheme in Fig. 1? Same goes for

addition of SB-43154 in line 102. in lines 106 to 108: The authors should also state the specific time point of 50% induction efficiency.

The authors need to clarify the time point when cultures were assessed with specific experiments. For example, in Fig. 1 the exact time points of flow cytometry assessment of the cultures and performance of real time qPCR experiment in G should be specified.

We apologize for not being clear and have now added additional language and improved our schematics and figure legends to show exactly which factors are being tested, when they are added, and when the cells are analyzed.

Revision:

- Throughout the manuscript, Figures, and Figure Legends

6. In many cases, the poor resolution of the images makes it difficult to make out which genes were analyzed in 2D. These genes are separated into tendon, proliferating, and cartilage genes. Did the authors test for the presence of genes for bone, ligament, muscle, adipose markers or any other musculoskeletal genes?

We apologize for the poor resolution. We have now double checked this issue and hopefully the images are now visually accessible.

To address the reviewer's point on other lineages, we now include more data in the results discussing non-tendon/ligament markers that were identified in the scRNA-seq, which is the best way to determine heterogeneity of the differentiated culture. This is included as a dot plot in the new Supplementary Figure 4. We now also list the percentage contribution of the cultured cells to these non-connective tissue lineages (which was defined using E14.5 tail cells) in the new Table 1. For the most part, these cells represented a small fraction of the culture.

In the new results for the 3D engineered tissues, we also now report bone, muscle, and adipose markers in comparison with 2D cultures. We did not find enhanced expression of these markers with 3D culture.

Revision:

- New data in Supplementary Figure 4, Table 1 (Pg 37), Supplementary Figure 9
- New results described in Pg 9, 15-16

7. Lines 329-331: If manipulating BMP signaling did not affect ScxGFP induction within this model, how does that affect the cells the authors are analyzing? The results seem questionable if these experiments cannot in some way replicate well known tendon research from other papers.

The role of BMP signaling on Scx expression and tendon induction was initially thought to be purely repressive, based on early studies in the chick limb (Schweitzer et al., 2001). While at early stages in mouse limb development, BMP4 signaling also directs tendo-chondro progenitors toward the cartilage fate (Blitz et al., 2009), a new study from Dr. Schweitzer's lab (Dr. Huang is also a coauthor) now shows that tendons express several BMPs (including BMP2, BMP7, and BMP4) as well as pSmad1/5/8, and that tendon-specific deletion of BMP signaling either through Smad4 or BMP receptors ALK3/ALK6 results in a tendon growth and maturation

phenotype suggesting that the role of BMP signaling in tendon fate is actually quite complex and not purely inhibitory (Schlesinger et al., 2021). These points are now included in the discussion.

To directly determine the effect of BMP signaling on tendon differentiation in our system, we now include new data testing BMP4 alone or in combination with TGFB1. We also test whether inhibition of BMP signaling by LDN193189 in combination with TGFB1 will improve induction efficiency. Our results show that while BMP4 treatment alone completely inhibited ScxGFP induction, this was not the case in the presence of TGFB1. Interestingly, inclusion of LDN actually slightly reduced efficiency indicating a requirement for BMP signaling in Scx induction, but only in the presence of TGFB.

Revision:

- New data shown in Supplementary Figure 2
- New results described in Pg 7
- Additional discussion in Pg 18-19

8. It is unclear how the authors are harvesting mESCs from transgenic ScxGFP mice. Scleraxis in developing mice is a marker for all connective tissue including tendons as well as ligaments. The authors isolated ScxGFP mESCs (which is not described) and then used the addition of various factors to initiate tenogenic differentiation. How do the authors control for differentiation into other cell lines? What other markers did they test to make sure that their cultures are relatively 'pure' tendon progenitors and not expressing markers for other connective tissues like ligament? In vitro differentiation can result in unintended differentiation into other lines. What sort of quality control did these cultures undergo to make sure that cultures were relatively 'pure'?

We have now described mESC derivation in the methods. To clarify, ScxGFP is not expressed at the blastocyst stage (~E3.5), when mESCs are derived from the inner cell mass. We obtained 13 lines and genotyped for ScxGFP to determine which lines contained the transgene. We also carried out all experiments using a ScxGFP-negative line derived from the same litter, to use to gate ScxGFP in flow cytometry.

To determine differentiation into other lineages, we now include additional data from scRNA-seq (as also described in response to critique #6) to better identify the non-Scx expressing clusters which would indicate alternative phenotypes as well as the percentage of these cells in the cultured population. To date, scRNA-seq is the best validation of culture heterogeneity since each individual cell can be queried (vs bulk RNA-seq or qPCR where all cells are combined). This data is now presented and discussed in the Results as Table 1 and Supplementary Figure 4. Note that no differentiation strategy is ever 100% efficient. In other fields for example, the best efficiencies can range from 13% (ie notochord, (Zhang et al., 2020) up to 65-80% (ie cardiomyocytes or skeletal muscle, (Chal & Pourquie, 2017; Loh et al., 2016).

Revision:

- Updated methods Pg 22
- New data in Supplementary Figure 4, Table 1 (Pg 37)
- New results described in Pg 9

9. Fig. 1G. Authors may include the controls of undifferentiated mESC. This is important for understanding the level changes during differentiation.

We now include qPCR analysis of pluripotent markers that are high in mESCs and downregulated with differentiation. We also show that Scx is low in both mESC and PM cultures, but high after tenogenic differentiation (Supplementary Figure 1).

Revisions:

- New data shown in Supplementary Figure 1
- New results described in Pg 6

10. Fig. 4. The specific markers used in Fig. 3 should be repeated in Fig. 4.

Revisions:

- This is now shown in Supplementary Figure 6 and results in Pg 12-14

11. Fig. 3I. The gene expression is only investigated at a single time point at the final stage. More time points may be tested for obtaining the gene changes so that differentiation process can be better defined.

We agree. While the scRNA-seq was only carried out at the terminal timepoint (D10), we also included temporal trajectory analyses using RNA-seq. Cells were sampled for RNA-seq at multiple timepoints starting from the D6 PM stage thru D10 differentiation. From this we were able to identify transcriptional modules and signatures associated with differentiation. We also now include new data exploring additional timepoints, including the effect of extended passage and 2D culture (up to D18). We also show the results of extended 3D culture (D17).

Revisions:

- In addition to original RNA-seq data (now Figures 4 and 5), new data is shown in Supplementary Figure 8, Figure 6
- New results described in Pg 14-16

Response to Reviewer 2 Comments

1. However, I consider the overall innovation of this study slightly insufficient. The authors apply widely used transcription factors to induce ESCs that are few studied in this research area and the results do fill in the blanks.

We hope our new results and clarifications have now improved reviewer enthusiasm for the innovation. We believe this will be highly cited as the most comprehensive analyses of tendon/fibrocartilage cell fate induction to date and provide the developmental biology and tissue engineering fields with new cell fate markers and critical transcriptional data to move these fields forward. In addition, while TGF β signaling is widely studied for tendon/fibrocartilage, our results show intriguing and unexpected interactions and modifying

effects between TGF β , BMP, hedgehog, and retinoic acid signaling. Such studies are very difficult to carry out *in vivo*, but can be directly tested *in vitro*.

Revisions:

- New data shown in Figure 6, Figure 7, Supplementary Figures 1, 2, 4, 5, 6, 7-9, Table 1
- New results described in Pg 6-7, 9, 11-12, 14-17
- Additional discussion in Pg 18-22

2. While ESCs also have limitation in source of acquisition and transplantation for regeneration and repair *in vivo* (Lui PP, Stem Cells Cloning. 2015; 8:163-174. / Shojaee A, et al. Stem Cell Res Ther. 2019; 10(1):181.) compared to other previous researches using other kinds of stem cells as mentioned in the manuscript.

We completely agree with the reviewer's point that there are significant concerns (ethically and practically) with using ESC-derived cells for transplantation and therapeutic studies and have included the helpful reference from Dr. Lui to the discussion. Indeed, within our scRNA-seq data, we identified a population of undifferentiated cells that still expressed pluripotency markers (Supp Fig 3), which may give rise to problematic teratomas if implanted. As discussed in this interesting perspective by Shinya Yamanaka (Yamanaka, 2020), one way to avoid the important issues identified by the reviewer is to improve differentiation efficiency. We therefore addressed this rogue population by improving our paraxial mesoderm differentiation efficiency (and subsequent tendon/fibrocartilage efficiency) to 90% through early application of AGN.

While this concern can also be addressed by simply sorting ScxGFP cells (the incorporation of the reporter is a strength of these studies) at the end of differentiation, we ultimately believe a better application of our platform is for screening, disease modeling, and determining cell fate decisions *in vitro* to advance scientific understanding. These types of applications have high significance both for developmental and regenerative processes. Indeed, ESCs are still widely used by many of the major stem cell labs for these purposes (for example Gordon Keller, Austin Smith, Olivier Pourquie and Irv Weissman to name a few) and this work continues to be very impactful. In Cell Stem Cells in 2020 alone, several studies from purely *in vitro* systems were published (Gage et al., 2020; Hendriks, Clevers, & Artegiani, 2020; Ortmann et al., 2020; Skelly et al., 2020; Yang et al., 2020; Yeo et al., 2020; Yilmaz, Braverman-Gross, Bialer-Tsypin, Peretz, & Benvenisty, 2020).

Revisions:

- Additional discussion in Pg 18

3. Besides, the authors culture these cells under 2-dimensional environment and generate tendon and fibrocartilage cells with transcriptional signatures comparable to *in vivo* embryonic cells. As many previous studies use 3-dimensional culture to induce tendon cells (Chen X, et al. Sci Rep. 2012; 2:977. / Theodossiou SK, Schiele NR. BMC Biomed Eng. 2019; 1:10.1186.) and some indicate 3-dimensional culture may be more suitable for induction and closer to the environment *in vivo* (Barsby T, et al. Tissue Eng Part A. 2014; 20(19-20):2604-2613. / Bavin EP, et al. Front Vet Sci. 2015;2:55.). Based on the above two points, I consider this model has certain limitation in wide application.

We thank the reviewer for this important insight and have now included new data confirming that extended 2D culture results in loss of tendon phenotype. We also generated 3D engineered tissues to address this point and compared to 2D culture. Finally, we adapted and validated a bioreactor system to demonstrate the utility of the system for studies of tendon mechanobiology during development. We believe these additional experiments have tremendously improved the impact and relevance of the paper and appreciate the reviewer's comments.

Revision:

- New data shown in Figure 6, Figure 7, Supplementary Figure 8, Supplementary Figure 9
- New results described in Pg 14-17
- Additional discussion in Pg 21-22

4. Also, it would be more convincing and comprehensive if the authors add some other experiment results in vivo and vitro (such as some morphological results, proteomics experiments, repairation results and so on. Komura S, et al. Sci Rep. 2020; 10(1):3992. / Tan GK, et al. Elife. 2020; 9: e52695.)

While we considered conducting repair studies using the cells, the point made by the reviewer (that mESC-derived cells are ultimately not a clinically-translational source of repair cells) turned our efforts in a different direction. Since our intention was to model development, we now include new data collected from in vivo embryonic tissues from tendon induction through differentiation stages (E11.5-E14.5). To address the reviewer's point regarding proteomics, we also assessed induction of other signaling pathways using a proteomic panel for a broad range of phosphorylated receptor tyrosine kinases and found increased phosphorylation of RTKs associated with chondrogenesis in the fibrocartilage condition. Finally, we provide morphological data using multiphoton imaging of cells in the 3D engineered tissues to show alignment along the direction of tension and second harmonic generation imaging to show deposition of aligned collagen.

Revision:

- New data shown in Figure 6, Supplementary Figure 5, Supplementary Figure 7
- New results described in Pg 11, 14-16

5. In addition, some figures' resolution in the paper is too low for readers to figure out the words (fig. 2-4) and these figures could also use some aesthetical polishing.

We apologize greatly for this and have now fixed this issue. We have also increased text size for key panels and attempted to improve appearance of figures and organization.

Revision:

- All figures

Response to Reviewer 3 Comments

1. First, concerning the choreography of the TGF β and hedgehog pathways, how does the critical retinoic acid timing stay on schedule in vivo, with a more complicated circulation of factors such as TGF β ? Do the authors have any insight into how these detailed and apparently kinetics are activated in parallel for successful differentiation in development?

This is a great question and to my knowledge has not been addressed. In vivo, deletion of various retinoic acid receptors lead to defects in somitogenesis and paraxial mesoderm specification, as well as cartilage differentiation (Duester, 2007; Mendelsohn, Ruberte, & Chambon, 1992; Williams et al., 2009). Most of the RAR cartilage studies focus on long bone and growth plate development. It was shown that RARgamma in particular is critical for maintenance of cartilage markers, but only in the absence of ligand. This is quite consistent with our observation of RARgamma expression in the fibrocartilage cluster and how addition of agonist inhibits chondrogenesis. On the other hand, the major phenotype of TGF β signaling mutants appear to be related to loss of tendons, loss of the secondary bony structures (such as tuberosities), and fusion of selected joints. I don't believe RAR signaling has been studied in these contexts. In the literature, the data on TGF β /RA signaling suggests that the interaction between these pathways are highly context- and tissue/cell-dependent and RA can have activating or suppressive effects on TGF β signaling (and vice versa!)(Pendaries, Verrecchia, Michel, & Mauviel, 2003; Xu & Kopp, 2012). Most interestingly, for cartilage, RA binding to RAR results in suppression of Smad2/3 phosphorylation and reduced chondrogenesis. Taken together with recent data showing that Smad4 and Smad2/3 mutants do not recapitulate the tendon phenotypes of TGFBR2/ScxCre mutants, we posit that the tendon fate activities of TGF β signaling may be Smad-independent (Schlesinger et al., 2021; Tan et al., 2020). Thus the addition of RARgamma agonist CD1530 would perhaps suppress Smad-mediated chondrogenesis without impacting tenogenesis. We have now included some of these points in the Discussion.

Revision:

- Additional discussion in Pg 19-20

2. The second, related question relates to mechanical cues. TGF β is a highly mechanosensitive protein. Do the authors have any insight into how the cells, especially the fibrocartilaginous cells, are able to ignore their mechanical environments to reach the desired endpoints? The groups of Stavros Thomopoulos, Helen Lu, Johnna Temenoff, and many others who are interested in growing both tendon and the fibrocartilaginous tendon-to-bone insertion put much stock in the need for mechanical loading to drive cells to the desired phenotype in tissue engineered constructs, as do those of us who model the problem. Are all of these mechanical treatments simply ways of mechanically driving the TGF β and hedgehog choreography that the authors have characterized? The reviewer does not expect the authors to have definitive answers to these questions, but if they have insight the reviewer would value their thoughts on these matters in the discussion.

We were also curious about the effect of the mechanical environment vs the effect of chemical cues and now directly test this by embedding cells at the D6 PM stage into 3D tensioned

collagen gels and inducing tendon/fibrocartilage differentiation in this environment. We find that over a period of 11 days, the cells contract to form a linearized construct. Surprisingly, while we expected this environment to promote tenogenic induction at the expense of chondrogenesis, we observed very distinctive cartilage nodules within the AGN condition. This suggests that the chemical cues alone may be sufficient to overcome the mechanical cues, at least at early differentiation stages. It would be very interesting to create a bone-tendon unit and determine whether there is sequestration of these molecules at the interface associated with mechanical loading. We have expanded the section on mechanical cues in the Discussion.

Revision:

- New data shown in Figure 6, Figure 7, Supplementary Figure 9
- New results described in Pg 15-17
- Additional discussion in Pg 21-22

3. The final question relates to the growth plate. Given the remarkable switching that the authors have found, how do cells in the tendon-to-bone insertion choose and maintain their phenotype? With fibrocartilaginous and tendinous cells almost abutting one another, how do they shield one another? Modelers of the problem assert that mechanics locks in the fate niche, but the authors appear to have overturned this by showing that the fate can be determined independent of mechanics. How does it work instead?

As the reviewer knows well, the enthesis is formed by Gli1-lineage cells that are specified in the embryo by E14.5, and localized between the tendon and skeleton. We can imagine a scenario whereby Gli1 cells express ligands (or RARgamma receptor) and there are molecular gradients formed. There may also be opposing molecular gradients emanating from the bone as well as from the tendon that result in upregulation of antagonists or negative regulators. There may even be cues emanating from the muscle that might play a role; as the tendon elongates, then the signals from muscle may no longer be experienced by Gli1 cells and this may encourage a fibrocartilage phenotype.

4. The manuscript is very well written and the figures are beautiful. Very minor suggestions to consider:

(1) In figure 4, the reviewer would have rather seen identical ordinate axes on all graphs. If there is a reason not to do this, a word on what is gained from having the different ordinate axes in the discussion would be appreciated as the reviewer was not sufficiently refined to pick it up on his own.

We think the reviewer is asking here why the same y-axis range is not kept consistent for all of the graphs (apologies if we mis-interpreted). The axes were chosen to best show the patterns we identified in the modules (upregulation or downregulation with differentiation, transient expression patterns, fate-specific trajectories). We hope these graphs highlight these differences from the starting PM population. There is some interesting work on signaling pathways (Wnt, TGFb, NFkB, etc) by Marc Kirschner and Uri Alon's groups that shows cells sense fold change in signals relative to background levels, rather than sensing absolute levels.

5. (2) Figure 4 has so much interesting information that it was in places unreadable. Because this journal allows more than four figures, would it make sense to split the figure into two separate figures that span two journal pages?

We are happy the reviewer found Fig 4 informative and agree that it is too dense. We have divided the figure as suggested.

Revision:

- Figure 4, Figure 5

References Cited

- Blitz, E., Viukov, S., Sharir, A., Schwartz, Y., Galloway, J., Pryce, B., . . . Zelzer, E. (2009). Bone ridge patterning during musculoskeletal assembly is mediated through SCX regulation of Bmp4 at the tendon-skeleton junction. *Developmental cell*, 17(6), 861-873. doi:10.1016/j.devcel.2009.10.010
- Chal, J., & Pourquie, O. (2017). Making muscle: skeletal myogenesis in vivo and in vitro. *Development*, 144(12), 2104-2122. doi:10.1242/dev.151035
- Comai, G. E., Tesarova, M., Dupe, V., Rhinn, M., Vallecillo-Garcia, P., da Silva, F., . . . Tajbakhsh, S. (2020). Local retinoic acid signaling directs emergence of the extraocular muscle functional unit. *PLoS Biol*, 18(11), e3000902. doi:10.1371/journal.pbio.3000902
- Duester, G. (2007). Retinoic acid regulation of the somitogenesis clock. *Birth Defects Res C Embryo Today*, 81(2), 84-92. doi:10.1002/bdrc.20092
- Gage, B. K., Liu, J. C., Innes, B. T., MacParland, S. A., McGilvray, I. D., Bader, G. D., & Keller, G. M. (2020). Generation of Functional Liver Sinusoidal Endothelial Cells from Human Pluripotent Stem-Cell-Derived Venous Angioblasts. *Cell Stem Cell*, 27(2), 254-269 e259. doi:10.1016/j.stem.2020.06.007
- Giordani, L., He, G. J., Negroni, E., Sakai, H., Law, J. Y. C., Siu, M. M., . . . Le Grand, F. (2019). High-Dimensional Single-Cell Cartography Reveals Novel Skeletal Muscle-Resident Cell Populations. *Mol Cell*, 74(3), 609-621 e606. doi:10.1016/j.molcel.2019.02.026
- Hendriks, D., Clevers, H., & Artegiani, B. (2020). CRISPR-Cas Tools and Their Application in Genetic Engineering of Human Stem Cells and Organoids. *Cell Stem Cell*, 27(5), 705-731. doi:10.1016/j.stem.2020.10.014
- Irion, S., Nostro, M. C., Kattman, S. J., & Keller, G. M. (2008). Directed differentiation of pluripotent stem cells: from developmental biology to therapeutic applications. *Cold Spring Harb Symp Quant Biol*, 73, 101-110. doi:10.1101/sqb.2008.73.065
- Loh, K. M., Chen, A., Koh, P. W., Deng, T. Z., Sinha, R., Tsai, J. M., . . . Weissman, I. L. (2016). Mapping the Pairwise Choices Leading from Pluripotency to Human Bone, Heart, and Other Mesoderm Cell Types. *Cell*, 166(2), 451-467. doi:10.1016/j.cell.2016.06.011
- McGurk, P. D., Swartz, M. E., Chen, J. W., Galloway, J. L., & Eberhart, J. K. (2017). In vivo zebrafish morphogenesis shows Cyp26b1 promotes tendon condensation and musculoskeletal patterning in the embryonic jaw. *PLoS Genet*, 13(12), e1007112. doi:10.1371/journal.pgen.1007112
- Mendelsohn, C., Ruberte, E., & Chambon, P. (1992). Retinoid receptors in vertebrate limb development. *Dev Biol*, 152(1), 50-61. doi:10.1016/0012-1606(92)90155-a

- Ortmann, D., Brown, S., Czechanski, A., Aydin, S., Muraro, D., Huang, Y., . . . Vallier, L. (2020). Naive Pluripotent Stem Cells Exhibit Phenotypic Variability that Is Driven by Genetic Variation. *Cell Stem Cell*, 27(3), 470-481 e476. doi:10.1016/j.stem.2020.07.019
- Pendaries, V., Verrecchia, F., Michel, S., & Mauviel, A. (2003). Retinoic acid receptors interfere with the TGF-beta/Smad signaling pathway in a ligand-specific manner. *Oncogene*, 22(50), 8212-8220. doi:10.1038/sj.onc.1206913
- Schlesinger, S. Y., Seo, S., Pryce, B. A., Tufa, S. F., Keene, D. R., Huang, A. H., & Schweitzer, R. (2021). Loss of Smad4 in the scleraxis cell lineage results in postnatal joint contracture. *Dev Biol*, 470, 108-120. doi:10.1016/j.ydbio.2020.11.006
- Schweitzer, R., Chyung, J. H., Murtaugh, L. C., Brent, A. E., Rosen, V., Olson, E. N., . . . Tabin, C. J. (2001). Analysis of the tendon cell fate using Scleraxis, a specific marker for tendons and ligaments. *Development*, 128(19), 3855-3866. Retrieved from http://www.ncbi.nlm.nih.gov/entrez/query.fcgi?cmd=Retrieve&db=PubMed&dopt=Citation&list_uids=11585810
- Skelly, D. A., Czechanski, A., Byers, C., Aydin, S., Spruce, C., Olivier, C., . . . Reinholdt, L. G. (2020). Mapping the Effects of Genetic Variation on Chromatin State and Gene Expression Reveals Loci That Control Ground State Pluripotency. *Cell Stem Cell*, 27(3), 459-469 e458. doi:10.1016/j.stem.2020.07.005
- Tan, G. K., Pryce, B. A., Stabio, A., Brigande, J. V., Wang, C., Xia, Z., . . . Schweitzer, R. (2020). Tgfbeta signaling is critical for maintenance of the tendon cell fate. *Elife*, 9. doi:10.7554/eLife.52695
- Williams, J. A., Kondo, N., Okabe, T., Takeshita, N., Pilchak, D. M., Koyama, E., . . . Iwamoto, M. (2009). Retinoic acid receptors are required for skeletal growth, matrix homeostasis and growth plate function in postnatal mouse. *Dev Biol*, 328(2), 315-327. doi:10.1016/j.ydbio.2009.01.031
- Xu, Q., & Kopp, J. B. (2012). Retinoid and TGF-beta families: crosstalk in development, neoplasia, immunity, and tissue repair. *Semin Nephrol*, 32(3), 287-294. doi:10.1016/j.semnephrol.2012.04.008
- Yamanaka, S. (2020). Pluripotent Stem Cell-Based Cell Therapy-Promise and Challenges. *Cell Stem Cell*, 27(4), 523-531. doi:10.1016/j.stem.2020.09.014
- Yang, L., Han, Y., Nilsson-Payant, B. E., Gupta, V., Wang, P., Duan, X., . . . Chen, S. (2020). A Human Pluripotent Stem Cell-based Platform to Study SARS-CoV-2 Tropism and Model Virus Infection in Human Cells and Organoids. *Cell Stem Cell*, 27(1), 125-136 e127. doi:10.1016/j.stem.2020.06.015
- Yeo, G. H. T., Lin, L., Qi, C. Y., Cha, M., Gifford, D. K., & Sherwood, R. I. (2020). A Multiplexed Barcodelet Single-Cell RNA-Seq Approach Elucidates Combinatorial Signaling Pathways that Drive ESC Differentiation. *Cell Stem Cell*, 26(6), 938-950 e936. doi:10.1016/j.stem.2020.04.020
- Yilmaz, A., Braverman-Gross, C., Bialer-Tsypin, A., Peretz, M., & Benvenisty, N. (2020). Mapping Gene Circuits Essential for Germ Layer Differentiation via Loss-of-Function Screens in Haploid Human Embryonic Stem Cells. *Cell Stem Cell*, 27(4), 679-691 e676. doi:10.1016/j.stem.2020.06.023
- Zhang, Y., Zhang, Z., Chen, P., Ma, C. Y., Li, C., Au, T. Y. K., . . . Lian, Q. (2020). Directed Differentiation of Notochord-like and Nucleus Pulposus-like Cells Using Human Pluripotent Stem Cells. *Cell Rep*, 30(8), 2791-2806 e2795. doi:10.1016/j.celrep.2020.01.100

Reviewers' Comments:

Reviewer #1:

Remarks to the Author:

The authors should be lauded in that they are very responsive to reviewers' comments. Specifically, to address reviewers' questions, they have performed new experiments, added new results, and substantially revised the manuscript. As a result, this manuscript is greatly improved in its overall quality and information conveyed now is from more comprehensive data-sets. There are two questions, however, the authors may wish to clarify.

1. The description of RA on "Retinoic acid signaling regulates the switch between tendon and fibrocartilage cell fates."

Retinoic acid (RA) is a vitamin A metabolite that is essential for early embryonic development. It usually acts as an activator of undirected differentiation. This may mean that RA in this study triggers non-specific differentiation of ESCs and as a result, the differentiated cell population may contain different matured cells, such as neural cells, hemopoietic cells, epithelial cells, etc.. Thus, the differentiation switch of RA is limited within a specific type of cell differentiation, which usually requires a unique differentiation environment. The authors claim that D6 embryonic stem cells tend to generate more scx-GFP cells after differentiation with AGN193109+ RA treatment, whereas the treatment with RA+ATRA produces an opposite effect. It is known that RAR antagonist AGN 193109 may promote the differentiation of the primitive stem cells. However, as a different RAR agonist, ATRA actually enhances the maintenance and self-renewal of stem cells. The level of upregulated scx-GFP expression may only reflect the increase in undirected differentiation of ESCs by AGN, not a directed differentiation; also, the reduction of scx-GFP expression by ATRA may just indicate the inhibition of ESCs differentiation. In other words, RA treatment is necessary for undirected differentiation of ESCs, but not for a directed differentiation. If one takes a close look at D10 results, the gene expression of differentiated cells was obviously induced by TGF-beta treatment. Since D10 cells were already under treatment with specific differentiation agent like TGF-beta, inhibition of RAR by ATRA actually inhibited all directed differentiation. As a result, the tendon specific gene expression was also downregulated. On the other hand, AGN enhanced all directed differentiation, which could raise the level of tendon specific gene too. So, to exclude the effects by AGN on nonspecific differentiation, a key experiment should include a TGF-beta inhibitor as control with or without AGN193109 treatment to confirm the effects by RAR signaling pathway.

2. Figs. 6 and 7.

The authors have shown that CD1530 inhibited the collagen II expression in SCXGFP + cells and slightly increased other cartilage related gene expression. The results illustrated its opposite function to AGN193019. They should also include CD1530 to Fig. 3 experiments to identify whether CD1530 affected the differentiation of ESCs at D6 and D10. Both CD1530 and AGN193019 were shown to induce the differentiation of ESCs to scxGFP+ tendon cells. The difference between the two is that CD1530 inhibited the collagen II expression. The authors may include more tendon and fibrocartilage related genes to confirm this phenomenon. The said difference may just reflect the stages of differentiation under RAR signaling pathway, not a specific induced differentiation. CD1530 as a RAR-gamma antagonist has been shown to preserve tendon stem cell characteristics and inhibit heterotopic ossification. This indicates that CD1530 may just maintain the cell stemness; it does not act as a specific inhibitor of cartilage differentiation under potent TGF-beta treatment.

Overall, it is felt that as the data stand, there is no sufficient evidence yet in this study to establish that RAR signaling is "the switch" between tendon and fibrocartilage differentiation.

Reviewer #2:

Remarks to the Author:

This manuscript by Kaji et. al. applied scRNA-seq on tenogenic and fibro-chondrogenic induction cultures of mESC and mouse embryonic tail, in which they discovered novel factors and the critical role of retinoic acid pathway in the differentiation processes. They further validated the findings in both 2D and 3D culture models.

The reviewer thinks the revised manuscript has been greatly improved and the authors have well

answered the previous questions, especially those concerning 3D culture of tendon and fibrocartilage differentiation under mechanical tension. The slight lack of innovation in model and technique, as pointed by one of the previous reviewers, can be compensated with the finding of retinoic acid signaling in the switch between the two differentiative fates. Future studies may benefit from this one regarding efficient induction and maintenance of the two types of differentiation.

At this state, the manuscript is suggested to be published.

Reviewer #3:

Remarks to the Author:

The manuscript is much strengthened and now constitutes an important and compelling contribution to the literature.

The reviewer was particularly enthusiastic about the new schematic in Figure 8 and the new quantification in Figure 6.

Congratulations!

Guy Genin

Reviewer Comments

Reviewer 1:

The authors should be lauded in that they are very responsive to reviewers' comments. Specifically, to address reviewers' questions, they have performed new experiments, added new results, and substantially revised the manuscript. As a result, this manuscript is greatly improved in its overall quality and information conveyed now is from more comprehensive data-sets.

Reviewer 2:

This manuscript by Kaji et. al. applied scRNA-seq on tenogenic and fibro-chondrogenic induction cultures of mESC and mouse embryonic tail, in which they discovered novel factors and the critical role of retinoic acid pathway in the differentiation processes. They further validated the findings in both 2D and 3D culture models.

The reviewer thinks the revised manuscript has been greatly improved and the authors have well answered the previous questions, especially those concerning 3D culture of tendon and fibrocartilage differentiation under mechanical tension. The slight lack of innovation in model and technique, as pointed by one of the previous reviewers, can be compensated with the finding of retinoic acid signaling in the switch between the two differentiative fates. Future studies may benefit from this one regarding efficient induction and maintenance of the two types of differentiation.

At this state, the manuscript is suggested to be published.

Reviewer 3:

The manuscript is much strengthened and now constitutes an important and compelling contribution to the literature. The reviewer was particularly enthusiastic about the new schematic in Figure 8 and the new quantification in Figure 6. Congratulations!

Author Response: We thank the reviewers for appreciating the additional data which we also feel greatly enhanced the impact and rigor of these studies. We have now included some new data to address Reviewer 1's outstanding concerns related to the role of retinoic acid signaling in the differentiation protocols. Changes to the text and new data is indicated by highlighting in the main manuscript. The manuscript now includes 8 main figures, 1 table, 11 supplementary figures, and 5 supplementary tables.

Response to Reviewer 1 Comments:

The description of RA on "Retinoic acid signaling regulates the switch between tendon and fibrocartilage cell fates." Retinoic acid (RA) is a vitamin A metabolite that is essential for early embryonic development. It usually acts as an activator of undirected differentiation. This may mean that RA in this study triggers non-specific differentiation of ESCs and as a result, the differentiated cell population may contain different matured cells, such as neural cells, hemopoietic cells, epithelial cells, etc.. Thus, the differentiation switch of RA is limited within a specific type of cell differentiation, which usually requires a unique differentiation environment.

The authors claim that D6 embryonic stem cells tend to generate more scx-GFP cells after differentiation with AGN193109+ RA treatment, whereas the treatment with RA+ATRA produces an opposite effect. It is known that RAR antagonist AGN 193109 may promote the differentiation of the primitive stem cells. However, as a different RAR agonist, ATRA actually enhances the maintenance and self-renewal of stem cells. The level of upregulated scx-GFP expression may only reflect the increase in undirected differentiation of ESCs by AGN, not a directed differentiation; also, the reduction of scx-GFP expression by ATRA may just indicate the inhibition of ESCs differentiation.

In other words, RA treatment is necessary for undirected differentiation of ESCs, but not for a directed differentiation. If one takes a close look at D10 results, the gene expression of differentiated cells was obviously induced by TGF-beta treatment. Since D10 cells were already under treatment with specific differentiation agent like TGF-beta, inhibition of RAR by ATRA actually inhibited all directed differentiation. As a result, the tendon specific gene expression was also downregulated. On the other hand, AGN enhanced all directed differentiation, which could raise the level of tendon specific gene too. So, to exclude the effects by AGN on nonspecific differentiation, a key experiment should include a TGF-beta inhibitor as control with or without AGN193109 treatment to confirm the effects by RAR signaling pathway.

The reviewer identified an interesting and important point about the different potential activities of RA signaling within the context of cell competence. We fully agree that RA signaling has multiple roles during development and that this is highly dependent on both timing of the factor and the state of cells receiving the signals. Not only does activation of RA signaling maintain pluripotency in ESCs (as the reviewer indicates), RA signaling may also direct direct epiblast cells toward a neural fate at the expense of paraxial mesoderm formation. We propose that either of these effects would be blocked by AGN treatment during D2-D6 stages, which would push the differentiation toward paraxial mesoderm specification. However, as the reviewer noted, we do not have direct evidence for this.

To better determine the identity of the D6 cells, we therefore carried out a new experiment, comparing ATRA and AGN treatment from D2-D6 and querying gene expression at D6 of the resulting cells. If RA signaling maintains stemness, we would expect as the reviewer indicates, that treatment with ATRA would maintain large numbers of undifferentiated cells that subsequently can give rise to other lineages (undirected differentiation). If ATRA activation of RA signaling on the other hand, is inducing neural formation, then the low numbers of ScxGFP cells at D10 can be explained by the fact that the D6 cells are no longer competent to undergo TGF β -induced tendon/fibrocartilage induction as they have already adopted an alternative fate.

While we expected D2-D6 ATRA treatment to maintain pluripotency, we were surprised to observe dramatic downregulation of Oct4, to comparable levels as D2-D6 AGN treatment. Analysis of epiblast markers (Eomes, Fgf5) showed minimal expression of these markers with ATRA, while the neural marker Sox1 was highly expressed compared to AGN treatment. Notably, paraxial mesoderm marker Tbx6 was highly expressed in AGN condition compared to ATRA. Collectively, these results indicate that while RA signaling did not maintain pluripotency or epiblast states, cells may have been pushed toward a neural fate at the expense of paraxial mesoderm by D6. Thus, the D6 cells induced by ATRA were no longer competent to undergo subsequent fibrous differentiation despite TGF β treatment at D6-10, thus explaining the reduced efficiency of ScxGFP induction at D10. We now include these interesting results

as new Supplemental Figure 5. We also checked our RNA-seq results for non-fibrous markers (such as for blood, nerve, muscle, and endoderm) and found minimal expression of these markers with either CD1530 or AGN treatment (see below). Note CPM values for these markers are quite low, suggesting minimal differentiation toward these fates with TGF β treatment, independent of RA signaling.

To the Reviewer's points about the role of RA signaling during the tenogenic induction phase (D6-10) - we completely agree that TGF β signaling is absolutely essential for tendon and fibrocartilage development, which was previously shown *in vivo*. In the absence of TGF β signaling, whether exogenously applied or endogenously generated (as we found in the FGF2-only or SAG-only conditions), the fibrous phenotype likely cannot be efficiently induced.

Therefore, we propose that TGF β induces the fibrous phenotype but RA signaling suppresses chondrogenic markers, without impacting the fibrous markers. Thus, RA modulation cannot compensate for loss of TGF β signaling as it cannot induce fibrous markers in the absence of TGF β . Intriguingly, during chondrocyte maturation, RA signaling is contextualized in a different manner when BMP is present (Li et al., 2003). Rather than suppressing chondrogenesis, the presence of RA induces chondrocyte hypertrophy. In limb bud cultures however, treatment with Vitamin A (to activate RA signaling) results in suppression of chondrogenesis while myogenesis was not affected (Pacifi, Cossu, Molinaro, & Tato, 1980). RA signaling therefore induces different responses depending on cell type. We have now provided a schematic to show the role of RA and TGF β signaling within the context of our differentiations. We hope this clarifies the different temporal activities for RA signaling during fate specification. We have also included more careful language in the discussion clarifying the role of RA signaling in fate switch is in the context of TGF β signaling.

Revision

- New data in Supplementary Figure 5
- New schematic clarifying the role of RA signaling during paraxial mesoderm differentiation and during tendon/fibrocartilage induction (Supplementary Figure 7)

- Revised text in abstract (pg. 2), introduction (pg. 4), results (pgs. 11-12), discussion (pgs. 20-21)

2. Figs. 6 and 7. The authors have shown that CD1530 inhibited the collagen II expression in SCXGFP + cells and slightly increased other cartilage related gene expression. The results illustrated its opposite function to AGN193019. They should also include CD1530 to Fig. 3 experiments to identify whether CD1530 affected the differentiation of ESCs at D6 and D10. Both CD1530 and AGN193019 were shown to induce the differentiation of ESCs to scxGFP+ tendon cells. The difference between the two is that CD1530 inhibited the collagen II expression. The authors may include more tendon and fibrocartilage related genes to confirm this phenomenon. The said difference may just reflect the stages of differentiation under RAR signaling pathway, not a specific induced differentiation. CD1530 as a RAR-gamma antagonist has been shown to preserve tendon stem cell characteristics and inhibit heterotopic ossification. This indicates that CD1530 may just maintain the cell stemness; it does not act as a specific inhibitor of cartilage differentiation under potent TGF-beta treatment.

Small note to clarify that increased cartilage expression is not observed with D6-D10 CD1530 treatment, rather cartilage markers are suppressed. As described above, the activity of AGN 193109 and CD1530 is very different in our protocol depending on when it is added and in the presence of other signaling molecules. We hope our new Supplemental Figure 7 now resolves some of the confusing aspects of RA signaling.

As the Reviewer suggests, we now include data testing CD1530 during PM induction (D2-6) and show suppression of ScxGFP at D10, albeit not to the same extent as ATRA treatment. This is likely due to the fact that CD1530 is specific to RAR γ while ATRA targets all of the RAR receptors. However, a mere 15% ScxGFP induction does suggest that the majority of RA signaling during the D2-6 differentiation phase is likely RAR-gamma mediated. This data is now included as new Supplemental Figure 5.

The reviewer also references an interesting previous study showing RAR agonists (including CD1530) increase Scx expression in tendon stem cells derived from adult tendons while suppressing differentiation toward other lineages, including cartilage (Webb, Gabrelow, Pierce, Gibb, & Elliott, 2016). This is similar to our findings as well. Unlike our findings however, they also observed enhanced expression of pluripotency markers Oct4 and Ssea1 and suppression of tendon differentiation markers Col1a1, Fmod, and Col3a1. However, a close observation of the y-axes (Fig 4) shows remarkably low expression of Oct4 and Ssea1 in all conditions and a true pluripotent control cell source, such as ESCs, was not included. It is therefore difficult to determine whether this finding is biologically meaningful. Our results in Supplemental Figure 1 and the new Supplemental Figure 5 suggest that pluripotency markers are downregulated as mESCs exit pluripotency toward PM differentiation. Our results in Supplemental Figure 8 also shows strong upregulation of tendon markers Col1a1, Col5a1, Col3a1 and more with CD1530 treatment. Another difference between this study and ours is the effect of RA agonism on extended culture. We found that extended 2D culture with CD1530 did not maintain proliferation potential as cells largely stopped dividing after 2 passages. This further suggests that D6-D10 CD1530 treatment is not inducing a stem cell phenotype in the presence of TGF β . The difference may be due to adult vs embryonic cell differences or cell types (tendon stem cells vs paraxial mesoderm) used.

In terms of heterotopic ossification, RAR agonism is an exciting therapeutic for the treatment of HO. Indeed, Palovarotene (a RAR γ agonist), is currently in clinical trials for the genetic HO disease, FOP. HO

however, progresses through endochondral ossification and ectopic cartilage formation is the first step of HO. Elegant studies by Maurizio Pacifici and Masahiro Iwamoto showed that RA agonism targets this ectopic chondrogenesis and this is what prevents later differentiation toward bone (Shimono et al., 2011).

To summarize, we feel as the Reviewer does that the role of RA signaling is highly context-dependent and based on the competence of the cells receiving these signals. In the context of pluripotent stem cells, RA agonism maintains this stem cell state. In the context of more restricted progenitor cells however (such as differentiated mesoderm), RA agonism inhibits chondrogenesis. Within this later phase, the presence of other signaling pathways plays a critical role. We posit that the presence of TGF β signaling contextualizes RA signaling such that the fate-switch is restricted to tendon vs fibrocartilage. Our studies strongly suggest that during this latter phase of differentiation, a stem-like phenotype is not induced. These points have now been incorporated into the Discussion.

Revision

- New data in Supplementary Figure 5
- New schematic clarifying the role of RA signaling during paraxial mesoderm differentiation and during tendon/fibrocartilage induction (Supplementary Figure 7)
- Revised text in results (pgs. 10, 13), discussion (pgs. 20)

Overall, it is felt that as the data stand, there is no sufficient evidence yet in this study to establish that RAR signaling is “the switch” between tendon and fibrocartilage differentiation.

We hope the new data provided, updated results, and additional discussion addressing the Reviewer’s comments have now allayed these concerns.

References Cited:

- Li, X., Schwarz, E. M., Zuscik, M. J., Rosier, R. N., Ionescu, A. M., Puzas, J. E., . . . O’Keefe, R. J. (2003). Retinoic acid stimulates chondrocyte differentiation and enhances bone morphogenetic protein effects through induction of Smad1 and Smad5. *Endocrinology*, 144(6), 2514-2523. doi:10.1210/en.2002-220969
- Pacifici, M., Cossu, G., Molinaro, M., & Tato, F. (1980). Vitamin A inhibits chondrogenesis but not myogenesis. *Exp Cell Res*, 129(2), 469-474. doi:10.1016/0014-4827(80)90517-0
- Shimono, K., Tung, W. E., Macolino, C., Chi, A. H., Didizian, J. H., Mundy, C., . . . Iwamoto, M. (2011). Potent inhibition of heterotopic ossification by nuclear retinoic acid receptor-gamma agonists. *Nat Med*, 17(4), 454-460. doi:10.1038/nm.2334
- Webb, S., Gabrelow, C., Pierce, J., Gibb, E., & Elliott, J. (2016). Retinoic acid receptor signaling preserves tendon stem cell characteristics and prevents spontaneous differentiation in vitro. *Stem Cell Res Ther*, 7, 45. doi:10.1186/s13287-016-0306-3

Reviewers' Comments:

Reviewer #1:

Remarks to the Author:

The authors are very responsive to the reviewer's comments. They provided new data and clarified the RA function in the directed differentiation of tendon and fibrocartilage. They also interpreted their data in the context of previous findings on the role of the activation of HH signaling pathway and TGF- β expression in the regulation of tendon-bone insertion development. The discussion has been substantially revised to make the conclusion "less sweeping" as compared to the previous version, and it is now more aligned with their data. This work will provide useful information to researchers for further investigations in the molecular mechanisms of tendon/fibrocartilage developmental biology, particularly in terms of additional molecular regulators involved in tendon/fibrocartilage differentiation.

Reviewer #1 (Remarks to the Author):

The authors are very responsive to the reviewer's comments. They provided new data and clarified the RA function in the directed differentiation of tendon and fibrocartilage. They also interpreted their data in the context of previous findings on the role of the activation of HH signaling pathway and TGF- β expression in the regulation of tendon-bone insertion development. The discussion has been substantially revised to make the conclusion "less sweeping" as compared to the previous version, and it is now more aligned with their data. This work will provide useful information to researchers for further investigations in the molecular mechanisms of tendon/fibrocartilage developmental biology, particularly in terms of additional molecular regulators involved in tendon/fibrocartilage differentiation.

We thank the reviewer for their positive comments!